# Immune checkpoint blockade in triple negative breast cancer influenced by B cells through myeloid-derived suppressor cells

Alyssa Vito [1,2], Omar Salem[1,2], Nader El-Sayes[1,2], Ian P. MacFawn[3,4,5], Ana L. Portillo [1,2], Katy Milne[6],
Danielle Harrington[6], Ali A. Ashkar [1,2], Yonghong Wan[1,2], Samuel T. Workenhe [7], Brad H. Nelson[6,8,9],
Tullia C. Bruno[3,4,5] & Karen L. Mossman [1,2✉]

Triple negative breast cancer holds a dismal clinical outcome and as such, patients routinely undergo aggressive, highly toxic treatment regimens. Clinical trials for TNBC employing immune checkpoint blockade in combination with chemotherapy show modest prognostic benefit, but the percentage of patients that respond to treatment is low, and patients often succumb to relapsed disease. Here, we show that a combination immunotherapy platform utilizing low dose chemotherapy (FEC) combined with oncolytic virotherapy (oHSV-1) increases tumor-infiltrating lymphocytes, in otherwise immune-bare tumors, allowing 60% of mice to achieve durable tumor regression when treated with immune checkpoint blockade. Whole-tumor RNA sequencing of mice treated with FEC + oHSV-1 shows an upregulation of B cell receptor signaling pathways and depletion of B cells prior to the start of treatment in mice results in complete loss of therapeutic efficacy and expansion of myeloid-derived suppressor cells. Additionally, RNA sequencing data shows that FEC + oHSV-1 suppresses genes associated with myeloid-derived suppressor cells, a key population of cells that drive immune escape and mediate therapeutic resistance. These findings highlight the importance of tumor-infiltrating B cells as drivers of antitumor immunity and their potential role in the regulation of myeloid-derived suppressor cells.

[1] McMaster Immunology Research Centre, McMaster University, Hamilton, ON, Canada. [2] Department of Medicine, McMaster University, Hamilton, ON, Canada. [3] Department of Immunology, University of Pittsburgh School of Medicine, Pittsburgh, PA, USA. [4] Tumor Microenvironment Center, UPMC Hillman Cancer Center, Pittsburgh, PA, USA. [5] Cancer Immunology and Immunotherapy Program, UPMC Hillman Cancer Center, Pittsburgh, PA, USA. [6] Deeley Research Centre, BC Cancer, Victoria, BC, Canada. [7] Department of Pathobiology, Ontario Veterinary College, University of Guelph, Guelph, ON, Canada. [8] Department of Biochemistry and Microbiology, University of Victoria, Victoria, BC, Canada. [9] Department of Medical Genetics, University of British Columbia, Vancouver, BC, Canada. ✉email: mossk@mcmaster.ca

Triple-negative breast cancer (TNBC) is an aggressive disease with dismal clinical outcome due to limited therapeutic options[1,2]. Immune checkpoint blockade (CP), which limits inhibitory pathways on CD8[+] and CD4[+] tumor-infiltrating lymphocytes (TILs), has demonstrated unparalleled clinical success in a wide variety of cancer types. Antibodies blocking the PD1: PDL1 inhibitory axis aim to target PD1 on T cells and PDL1 on tumor cells. However, only ~9% of patients with advanced TNBC express PDL1 on tumor cells[3]. Recently approved therapies for TNBC are limited (PARPi, atezolizumab, and sacituzumab-govitecan) and only benefit 10–20% of patients treated[3]. For this reason, a deeper understanding of the immune landscape in TNBC tumors is required to develop novel, effective therapies and delineate prognostic biomarkers of disease.

Many clinical trials are currently underway using CP as a standalone, adjuvant or neoadjuvant therapy for various tumor types[4]. However, the percentage of patients that respond to CP as a monotherapy is low and is generally limited to cancers with a favorable mutational landscape and high rates of cell turnover, such as melanoma and non-small cell lung cancer. In contrast, most breast lesions establish a highly suppressive tumor microenvironment (TME) with high levels of myeloid-derived suppressor cells (MDSCs), which has been shown to directly correlate with disease stage and overall tumor burden[5,6]. Additionally, clinical studies have shown the extent of immune cell infiltration and activation in breast tumors to be a key indicator of long-term survival and a predictor of therapeutic efficacy[7–10]. The use of therapies that can decrease MDSCs, increase TILs, and induce potent immunogenic cell death (ICD) may allow immunogenically cold breast tumors to better respond to CP.

A phase III clinical trial has shown that chemotherapy works to enhance the antitumor activity of CP in metastatic TNBC patients[3]. However, this combination of chemotherapy and immunotherapy only confers a modest overall survival benefit[3]. These results further highlight the need for additional strategies to treat TNBC and to mediate intrinsic resistance mechanisms. To this end, a more thorough analysis of the immune environment in TNBC patients beyond CD8[+] TILs is necessary.

Oncolytic viruses (OVs) have risen as an interesting therapeutic option with widespread clinical application and safe toxicity profiles due to their ability to preferentially target cancer cells over normal healthy cells. We have developed herpesvirus-based OVs that target multiple tumor types[11–14] and identified a strong correlation between OV-mediated ICD and antitumor immunity[14–19]. Moreover, oncolytic HSV-1s, such as oHSV-1 dICP0[14], synergize with low-dose chemotherapy to overcome immune tolerance[14,16–18]. Our early studies used mitoxantrone, a well-characterized immunogenic agent[18,20–22]. However, mitoxantrone is not normally used for breast cancer therapy, decreasing the translational value of our findings. Using a chemotherapy cocktail that is routinely used for TNBC patients will allow for ease of translation and biological analysis of current clinical applications.

Additional to the need for improved therapeutic approaches for TNBC patients is the simultaneous need to better understand the foundational biology that allows some patients to respond to treatment while their phenotypically identical counterparts do not. Here, we focused on the development of clinically relevant therapies that can be used to induce the inflamed microenvironment needed for tumors to respond to CP. We have used our therapeutic platform as a mechanism to study and better understand the complex biological processes that mediate therapeutic responses, as well as the interactions of key immune cell populations within the TME. This approach facilitates translation of findings to the clinic to improve immunotherapy responses in solid tumors.

## Results

**FEC + oHSV-1 improves survival outcome in TNBC tumor-bearing mice.** While chemotherapies are often given in dose-dense cytotoxic regimens, many studies have described the immunostimulatory properties of various chemotherapeutics through the use of low-dose intervention[23–26]. Based on current clinical practice, two of the most commonly used chemotherapy cocktails for breast cancer patients are FEC (5-fluorouracil, epirubicin, and cyclophosphamide) and AC (doxorubicin and cyclophosphamide)[27]. In an effort to reverse translate these regimens from the clinic into murine models, we performed optimization studies to find dosing levels and schedules that showed no acute cytotoxicity to murine hosts, as determined through close monitoring of body weights and scoring (Supplementary Fig. 1). In an effort to mimic clinical dosing, the individual ratios of the drugs to one another were the same as commonly prescribed in the clinic. The therapeutic potential of each regimen was evaluated in combination with oHSV-1 (Fig. 1a) using the E0771 TNBC syngeneic murine model. While no monotherapy showed efficacy, the addition of oHSV-1 to the clinical FEC regimen (FEC + oHSV-1) delayed tumor growth in some mice (Fig. 1b) and significantly extended survival, with 10% of mice resulting in durable tumor regression (Fig. 1c). Since epirubicin and doxorubicin are chemical analogues of one another, the main difference between these two regimens is in the addition of 5-fluorouracil to the FEC regimen. Indeed, a single dose of 5-fluorouracil alone has been shown to promote an antitumor immune response[28].

**FEC + oHSV-1 sensitizes tumors to checkpoint blockade therapy.** Although TNBC tumors were previously thought to be immunologically cold, recent clinical studies have shown that they do indeed express various immunogenic markers, such as PD-L1[29]. However, the expression of these markers is low and it is not diffuse throughout the tumor, but rather clumped in focal areas limited to a small proportion of cancer cells[30]. Further, clinical trials have reported both the efficiency and necessity of combined therapeutic modalities (e.g., immunotherapy and chemotherapy), as TNBC patients often have low, short-lived responses to CP on its own[31]. Based on our preliminary studies, we hypothesized that FEC + oHSV-1 is capable of sensitizing tumors to CP by turning an immune-cold tumor into an immunogenic one. Survival studies were performed with the addition of dual CP targeting the non-redundant pathways of cytotoxic T-lymphocyte antigen 4 and programmed death ligand-1 (with anti-CTLA4 and anti-PD-L1 antibodies, respectively). While neither CP alone, nor in combination with FEC or oHSV-1, showed therapeutic efficacy, the combination of FEC + oHSV-1 + CP resulted in greatly improved responses with 60% of mice achieving durable tumor regression (Fig. 2a–c). Furthermore, mice that showed durable regression of tumors subsequently rejected the parental (E0771) tumor cells when re-challenged, suggesting that responding mice generated immune memory against the tumor (Fig. 2d). Interestingly, when this triple combination therapy was tried with singular administration of CP (anti-CTLA4 mAb or anti-PD-L1 mAb) no therapeutic efficacy was seen (Supplementary Fig. 2), suggesting that targeting both pathways simultaneously is beneficial in overcoming resistance mechanisms in this aggressive tumor model.

To identify pathways essential for therapeutic efficacy, cytokine analysis was performed on tumor lysates obtained from mice treated with either saline or FEC + oHSV-1 + CP (Fig. 3; full dataset shown in Supplementary Fig. 3). Mice treated with FEC + oHSV-1 + CP had significantly changed expression levels of cytokines related to myeloid cell differentiation and chemotaxis, macrophage activation, and inflammatory pathways.

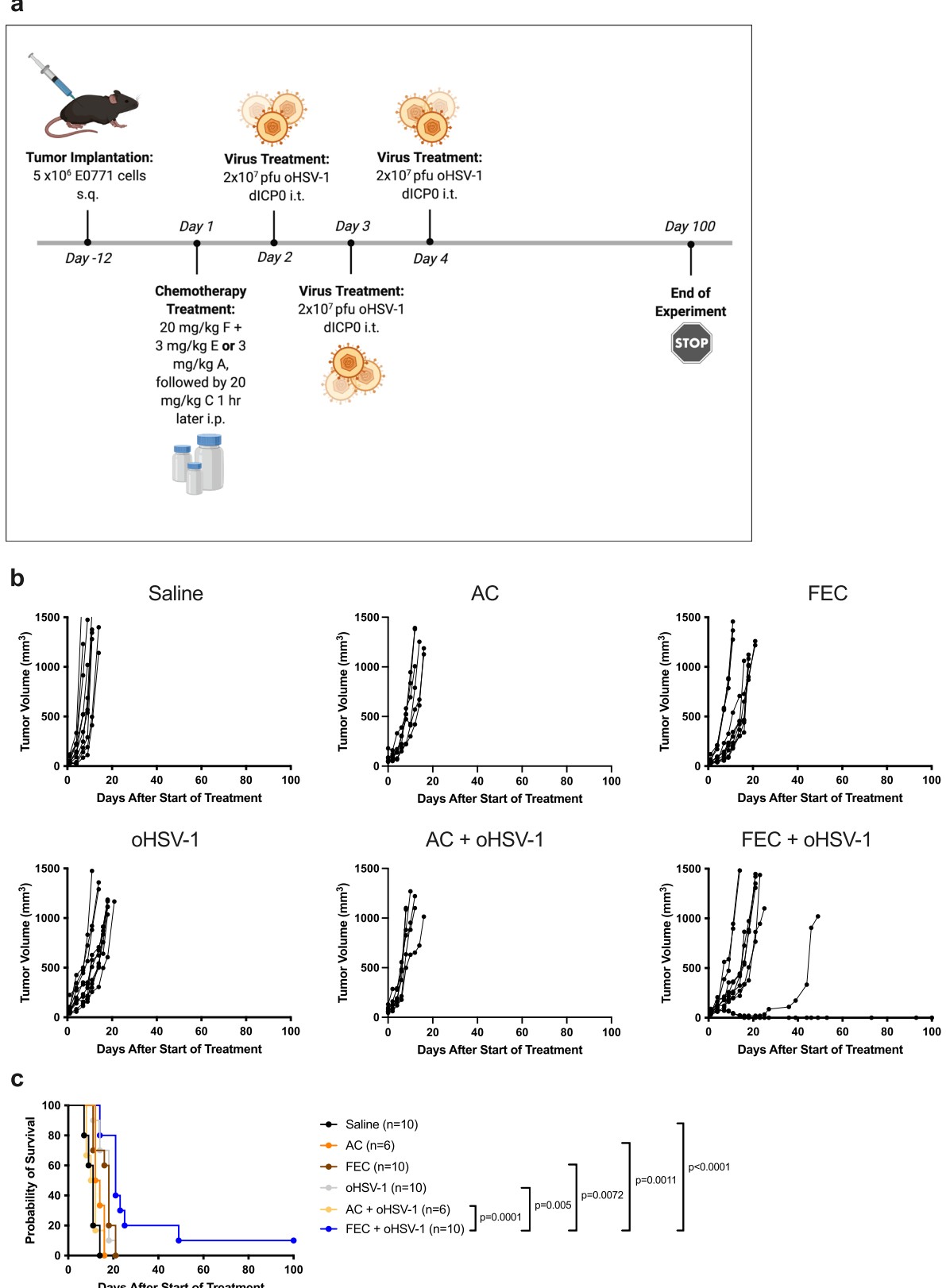

**Fig. 1 FEC + oHSV-1 slows tumor growth and decreases tumor kinetics and results in tumor regression in 10% of mice. a** C57/Bl6 mice bearing E0771 tumors were treated with saline, chemotherapy (FEC or AC), oncolytic virus (oHSV-1 dICP0), or chemotherapy + oncolytic virus. *Created using BioRender.com. **b** Tumor volumes were measured every 2–3 days from the start of treatment until mice reached endpoint. Each line represents an individual mouse within the group. **c** Kaplan–Meier survival curves of each group. *Mantel–Cox test was used for statistical analyses.

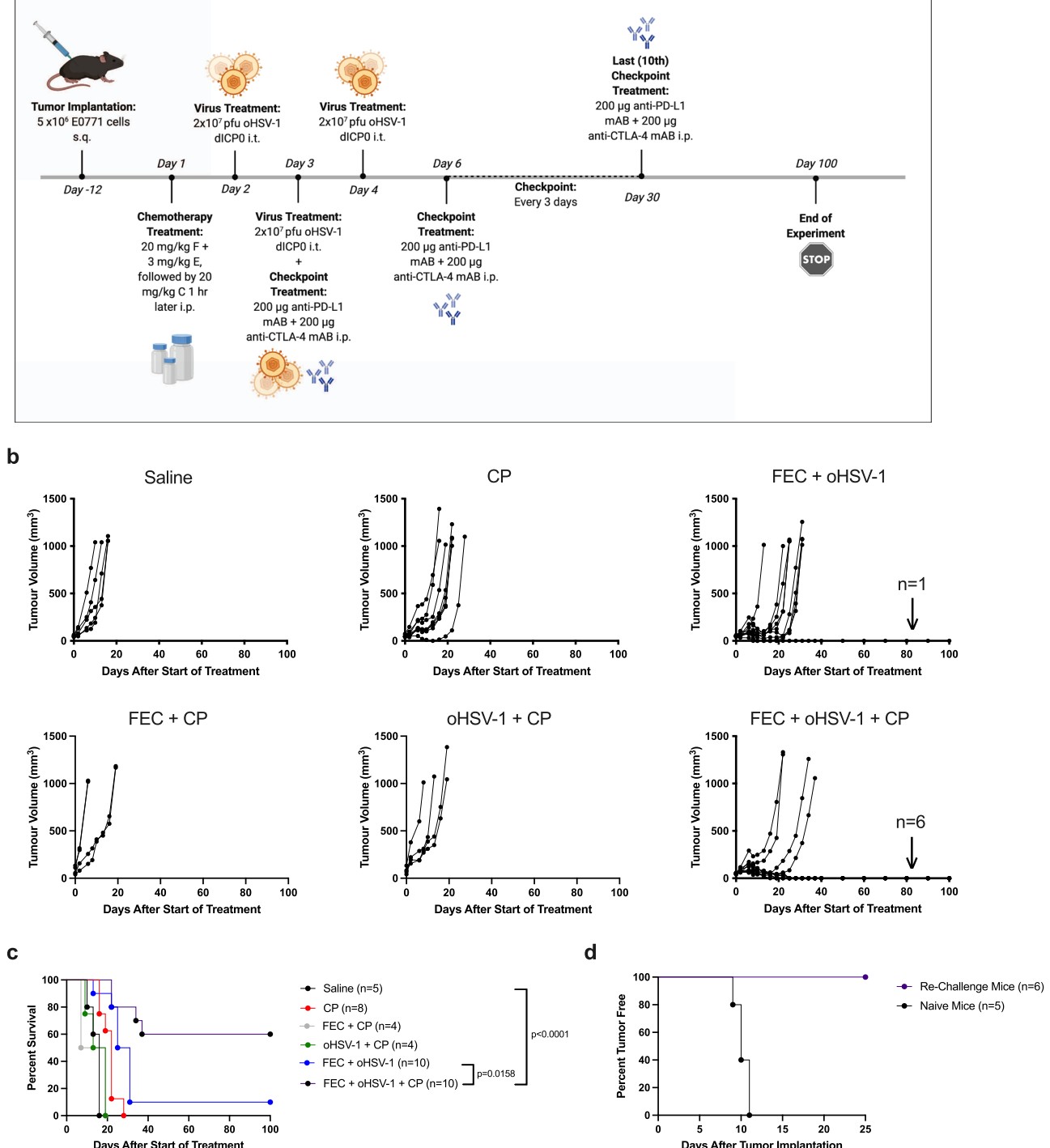

**Fig. 2 FEC + oHSV-1 + CP decreases tumor kinetics and results in durable tumor regression in 60% of mice. a** C57/Bl6 mice bearing E0771 tumors were treated with saline, CP (anti-CTLA4 and anti-PD-L1), FEC + CP, oHSV-1 + CP, FEC + oHSV-1, or FEC + oHSV-1 + CP. *Created using BioRender.com. **b** Tumor volumes were measured every 2–3 days from the start of treatment until mice reached endpoint. Each line represents an individual mouse within the group. **c** Kaplan–Meier survival curves of each group. **d** Mice who achieved durable tumor regression with FEC + oHSV-1 + CP therapy were subsequently re-challenged with E0771 cells on day 63 days. Naïve mice of similar age were used as a control for tumor implantation and growth. *Mantel–Cox test was used for statistical analyses.

**Immunogenic therapy induces TILs into otherwise immune-bare tumors**. To investigate the immune landscape of untreated and treated E0771 tumors, histologic assessment was performed. Tumors were harvested on days 7 and 10 from mice treated with saline, FEC, oHSV-1, FEC + oHSV-1, or FEC + oHSV-1 + CP

(Day 7, Fig. 4a; Day 10, Supplementary Fig. 4). Analysis of whole tumor sections harvested on day 7 and stained with hematoxylin and eosin (H&E) (Supplementary Fig. 5A) shows that saline and FEC-treated mice have large, densely packed tumors with many cells actively undergoing cellular division. Conversely, mice

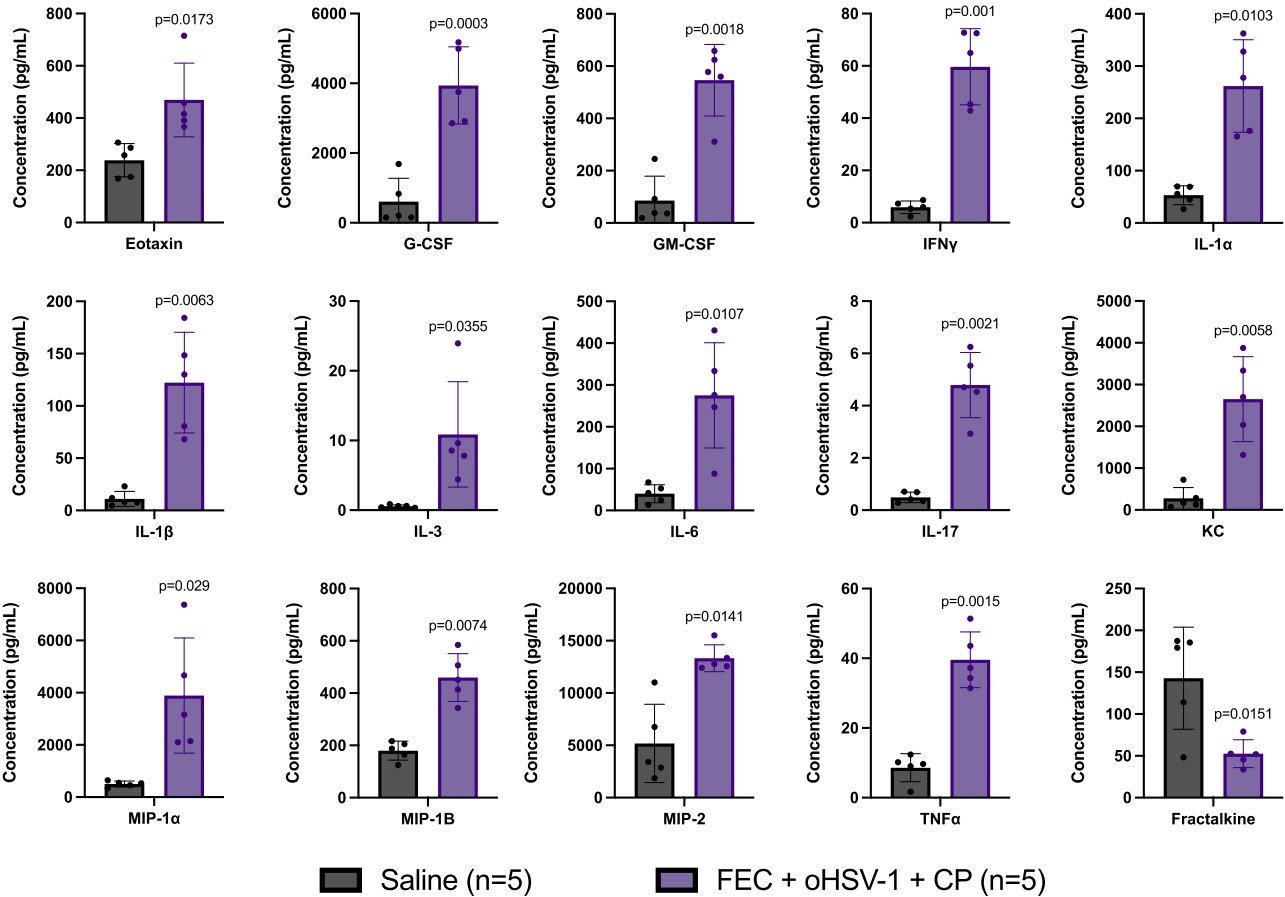

**Fig. 3 Cytokine analysis shows that treatment with FEC + oHSV-1 + CP significantly alters the expression of a variety of cytokines.** C57/Bl6 mice bearing E0771 tumors were treated with saline or FEC + oHSV-1 + CP. Mice were sacrificed on day 10 treatment and tumors flash frozen. Tumor lysates were made and sent for cytokines expression analysis. All values are reported as a concentration (pg/mL). Two-tailed paired *t* test was used for statistical analyses. Error bars are representative of the standard deviation.

treated with oHSV-1, FEC + oHSV-1, and FEC + oHSV-1 + CP present with shrinking cellular structures and large areas of necrosis, likely due to direct viral oncolysis from intratumoral administration of the virus. When these same groups are again analyzed on day 10 (Supplementary Fig. 5B) it is clear that saline and FEC-treated mice have continued to progress towards tumor endpoint with highly vascularized tumors undergoing extensive angiogenesis, displayed by the appearance of many new micro-vessels and infiltrating vasculature. oHSV-1-treated mice still have areas of necrosis in the center, but the remaining tumor tissue has begun to progress towards tumor endpoint, with large tumors that are densely packed with viable cells. Interestingly, FEC + oHSV-1-treated tumors now present two distinct popu-lations within mice, with 80% of tumors shifting towards an untreated phenotype with increased proliferation and density, similar in appearance to saline and FEC-treated tumors. The remaining 20% of tumors show increased areas of necrosis and decreased overall tumor size. As expected from survival outcomes, tumors harvested from mice that were treated with FEC + oHSV-1 + CP present with increased necrosis and very small tumors (Supplementary Fig. 5C).

Tumors were further stained with antibodies for CD3, CD4, and CD8α to assess the level of immune cell infiltration across the treatment groups (Fig. 4a). As expected, saline and FEC-treated mice presented with immune-bare tumors with almost no immune cells present in the tumor bulk or surrounding connective tissue. Tumors treated with oHSV-1, FEC + oHSV-1, or FEC +

oHSV-1 + CP, however, show increased levels of T-cell infiltration throughout the remaining viable tumor tissue. Quantification of these stains (Fig. 4b) reveals that tumors treated with oHSV-1, FEC + oHSV-1, or FEC + oHSV-1 + CP have statistically significant recruitment of immune cells into the tumor, as compared to saline-treated tumors. Clinical studies have shown the ratio of CD8+ T cells to FOXP3+ regulatory T cells to be a significant indicator of prognostic outcome in TNBC patients[32,33]. For this reason, all tumor slices stained for CD8α were co-stained for FOXP3 expression and the ratio between the two was assessed within individual mice (Fig. 4b). As shown, mice treated with oHSV-1, FEC + oHSV-1, or FEC + oHSV-1 + CP had statistically significant increases in their ratio of CD8α+/FOXP3+ infiltrates at day 7, but only with the combination therapies (FEC + oHSV-1 and FEC + oHSV-1 + CP) is this increase sustained out to day 10. This observation suggests that while oncolytic virotherapy is able to prime the tumor microenvironment, additional therapeutic inter-vention may be required for durable response to immunotherapy treatments.

**FEC + oHSV-1 induces a B-cell signature in whole tumor RNA sequencing.** Immunohistochemistry (IHC) clearly demonstrates that treatment with oHSV-1 creates an initial influx of T cells into the TME. However, IHC is unable to determine the functionality and activation state of these T cells and more specifically, whether they are capable of contributing to a robust antitumor immune response. The therapeutic efficacy of oHSV-1 as a monotherapy

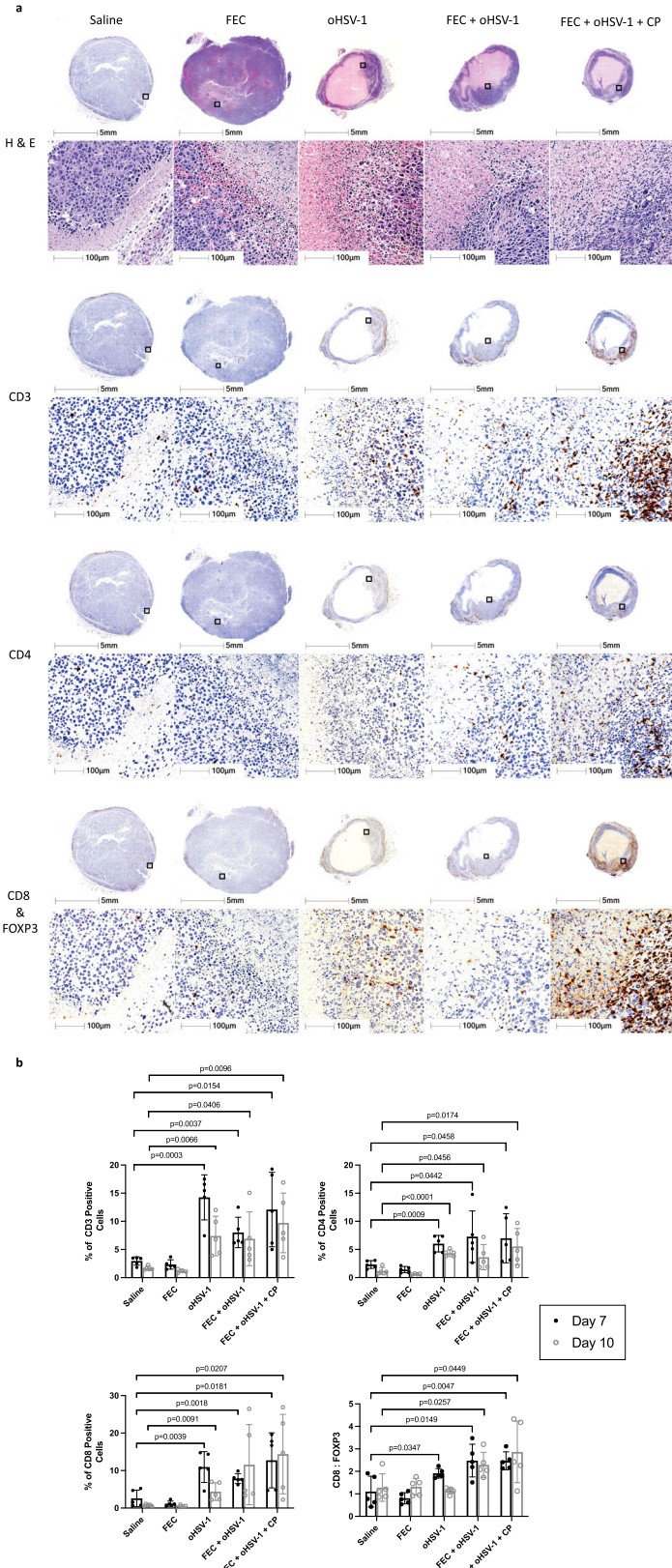

**Fig. 4 Immunohistochemistry analysis shows that oHSV-1, FEC + oHSV-1, and FEC + oHSV-1 + CP treatments induce TILs.** C57/Bl6 mice bearing E0771 tumors were treated with saline, FEC, oHSV-1, FEC + oHSV-1 or FEC + oHSV-1 + CP and tumors were harvested on day 7 ($n = 5$ per group). Tumors were sectioned and stained with H&E for pathological analysis. Sections were then further stained with antibodies for CD3, CD4, CD8α, and FOXP3. **a** Representative images for tumors harvested on day 7. Each image shows a whole section of an individual tumor. **b** Whole tumor sections were scanned and quantified using HALO quantification software. Each symbol represents an individual mouse within that group. Two-tailed paired $t$ test was used for statistical analyses. Error bars are representative of the standard deviation.

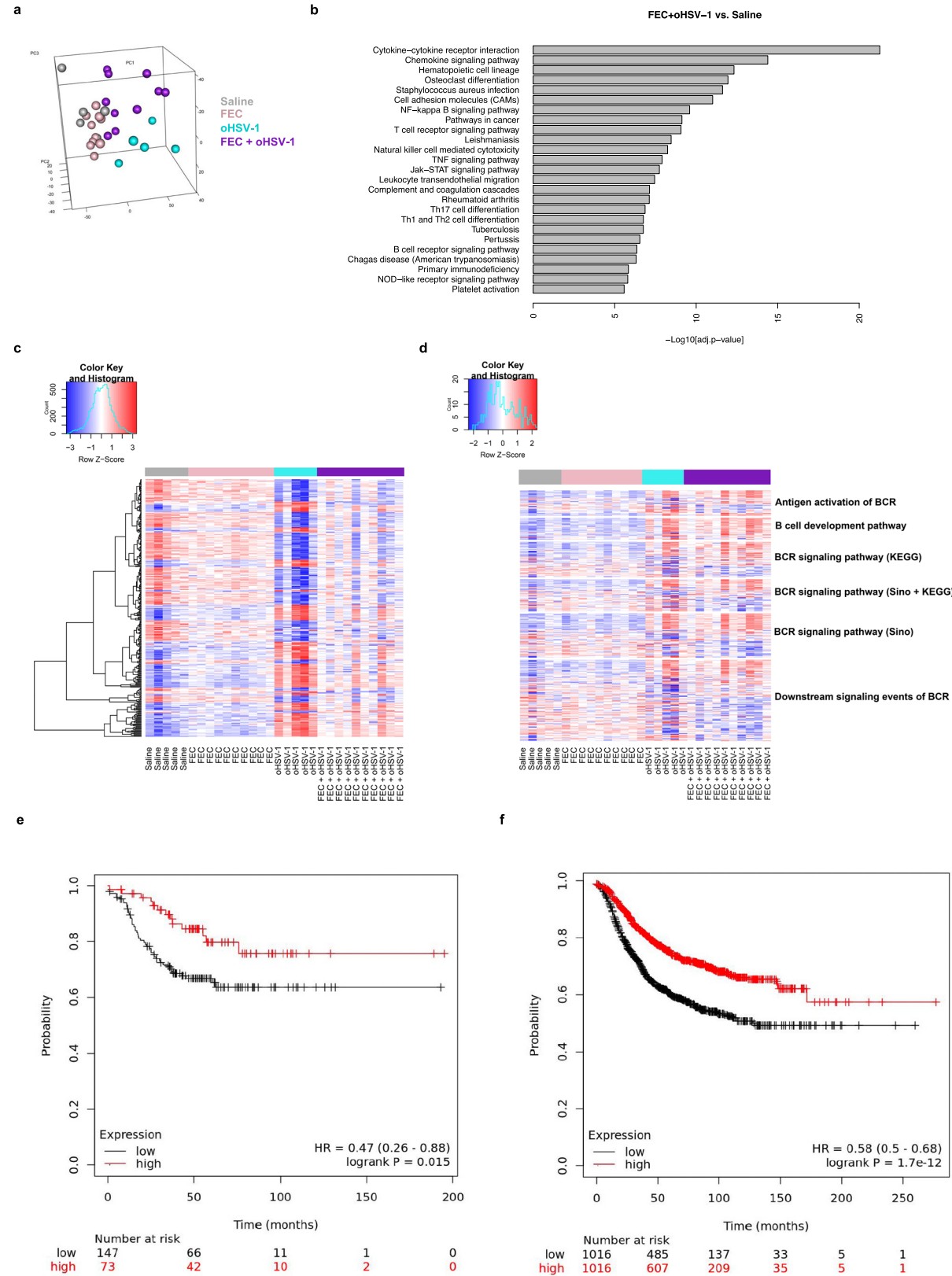

(as shown in Fig. 1c) shows that it is insufficient to limit tumor progression. To begin to understand how the addition of FEC mechanistically alters therapeutic responses to oHSV-1 treatment, we performed RNA sequencing analysis. Tumors were treated with saline, FEC, oHSV-1, or FEC + oHSV-1 and harvested on

day 5. Principal component analysis (PCA) shows that while saline and FEC-treated tumors cluster closely together, both oHSV-1 and FEC + oHSV-1 show distinct clustering patterns (Fig. 5a). Interestingly, within the FEC + oHSV-1 group we see two separate clusters of mice, one that has a similar expression

**Fig. 5 FEC + oHSV-1 induces significant upregulation in RNA transcriptomes associated with immune processes and pathways and more specifically, B-cell receptor signaling pathways. a** 3-D cluster plot showing the RNA expression correlations between mice treated with saline (gray; $n = 5$), FEC (pink; $n = 10$), oHSV-1 (teal; $n = 5$) and FEC + oHSV-1 (purple; $n = 10$). **b** Bar plot illustrating the results of pathway enrichment analysis performed on samples from mice treated with FEC + oHSV-1, compared to those treated with saline alone. **c** Heat map showing the normalized expression values of genes across all samples. **d** Heat map showing the normalized expression values of genes associated with B-cell receptor pathways. **e** Kaplan–Meier survival plot for TNBC patient RFS ($n = 220$) microarray data based on mean mRNA expression level of the top 29 available genes in Table S1. **f** Kaplan–Meier survival plot for all breast cancer patients RFS ($n = 2032$) based on mean mRNA expression levels of the top 29 available genes in Table S1. *BCR B-Cell Receptor.

profile to saline and FEC-treated mice (thought to be non-responders to treatment) and one that clusters distinctly (thought to be responders to treatment), consistent with previous data that FEC + oHSV-1 therapy results in a dichotomous response.

Pathway enrichment analysis identified many immune pathways and processes that were upregulated with FEC + oHSV-1 therapy (as compared to saline, Fig. 5b). Assessment of the differentially expressed gene pool (Fig. 5c) reveals that mice treated with FEC show similar genomic profiles to those treated with saline alone. However, treatment with either oHSV-1 or FEC + oHSV-1 shows a markedly different gene signature, with both oHSV-1 and FEC + oHSV-1 treatments upregulating many genes associated with the B-cell lineage (Fig. 5D, Supplementary Data 1), including *Ifitm1*, *Il1b*, *CD24a*, *CXCL12*, *FGF7*, and *Nrg1*. A signature comprising the top 29 upregulated genes from this analysis was surveyed across a large dataset of publicly available breast cancer patient data for prognostic relevance[34]. Breast cancer patients with mean expression of this signature in the highest tertile had significantly improved relapse-free survival (RFS) in TNBC [HR = 0.47 (0.26–0.86); logrank $p$ value = 0.015] (Fig. 5e), while mean expression above median across all patients was also correlated with improved RFS [HR = 0.58 (0.50–0.68); logrank $p$ value < 0.01] (Fig. 5f).

**Depletion of B cells results in loss of therapeutic efficacy.** While B cells play many different roles in the body, their primary function is in antibody production. To follow-up on the results of RNA sequencing analysis and to determine whether or not B cells play a key part in the efficacy of our combination therapies, we performed B-cell depletion studies. Subcutaneous E0771 tumors were grown in C57/Bl6 mice and circulating B cells were depleted using an anti-CD20 antibody (Fig. 6a).[35] Analysis of the overall survival shows that the depletion of B cells diminished therapeutic outcome in tumor-bearing mice (Fig. 6b, c, Supplementary Fig. 6). Depletion was confirmed via flow cytometry (Fig. 6d) and an isotype antibody was used as a control. Bulk antibody production levels were also assessed using an indirect ELISA assay. Serum was taken from naïve mice, tumor-bearing mice, and from therapeutically treated mice on day 15 and protein lysates from E0771 cells were used to coat the plates. Results of the ELISA assay indicates that FEC alone has no effect on antibody production levels. However, treatment with oHSV-1, CP, FEC + oHSV-1, and FEC + oHSV-1 + CP all show a statistically significant increase in antibody production levels, which is lower in the absence of circulating B cells (Fig. 6e). Since oHSV-1 or CP alone confer no therapeutic benefit, these data suggest that antibody production, while affected by our therapies, is not the most important function of B cells in relation to our therapeutic efficacy.

To further assess the role B cells play in our treatment, mice were injected with either an anti-CD20 antibody or an isotype antibody, treated with saline or FEC + oHSV-1 and tumors harvested on day 7. Quantification of IHC images shows that B-cell depletion results in decreased levels of tumor-infiltrating T cells and increased levels of Ly6G+ myeloid cells

(Supplementary Fig. 7A). Immunofluorescence (IF) staining was performed on whole tumor sections and stained for CD3, CD8, PNAd, Pax5, and CD11b. Analysis of multiplex images further corroborates the IHC quantification data and shows that mice treated with FEC + oHSV-1, in the presence of the isotype antibody, had formation of immature tertiary lymphoid structures (as evidenced by the presence of PNAd+ high endothelial venules) (Supplementary Fig. 7B). Conversely, mice treated with FEC + oHSV-1, in the presence of the anti-CD20 antibody, have no PNAd present and no organized lymphoid aggregates (Supplementary Fig. 7C).

**B cells are required for control of MDSCs and resultant therapeutic efficacy.** Depletion of circulating B cells results in a complete loss of therapeutic efficacy and disruption to immune cell organization in mice treated with FEC + oHSV-1 therapy. What remains unknown is how the addition of FEC mechanistically alters therapeutic outcome to oHSV-1 therapy and what effector cell functions are being regulated by tumor-infiltrating B cells (TIL-Bs). We performed a deeper assessment into the differentially expressed gene pool from RNA sequencing data, paying particular attention to the differences between oHSV-1 and FEC + oHSV-1 therapy. In this comparison we see many genes that are implicated in immunosuppressive MDSCs. Interestingly, some of these genes (*S100A8*, *CXCL2*, *CXCL1*, *Ly6G*, *Slpi*, *Fpr2*) are upregulated in oHSV-1 vs. saline but downregulated in FEC + oHSV-1 vs. oHSV-1 (Table 1, Fig. 7a), suggesting that FEC + oHSV-1 therapy is able to not only upregulate B-cell receptor signaling pathways, but also regulate the maturation and migration of MDSCs. It is well established in the literature that some chemotherapies (including 5-fluorouracil in particular) directly deplete MDSCs in both animal models as well as in patients[36–38]. However, in such studies chemotherapy is given in dose-dense cytotoxic regimens. In our hands, chemotherapy is being used as a low dose, immune-stimulatory intervention and shows no therapeutic efficacy alone, suggesting that it is unable to suppress MDSCs without the addition of oHSV-1.

To further investigate the relationship between these distinct cell types, we performed immune analysis studies. E0771 tumors were grown in C57/Bl6 mice and treated with saline, FEC, oHSV-1, CP, FEC + oHSV-1, or FEC + oHSV-1 + CP in both the presence (isotype mAb) and absence (anti-CD20 mAb) of circulating B cells. Blood was drawn on days 6, 10, and 15 and peripheral blood mononuclear cells (PBMCs) were analyzed via flow cytometry. While this analysis confirmed the depletion of circulating B cells in mice treated with our anti-CD20 mAB (Fig. 7b), it also revealed a striking and consistent difference in the population of MDSCs (Ly6G$^{hi}$Ly6C$^{int}$ cells) between mice treated with isotype and anti-CD20 antibodies (Fig. 7c). Analysis of the frequency of these populations shows that circulating B cells in isotype-treated mice decrease over time in all treatment groups, except for those mice treated with FEC + oHSV-1 + CP (Fig. 7d). On the contrary, mice treated with the triple combination therapy see a re-population of circulating B cells

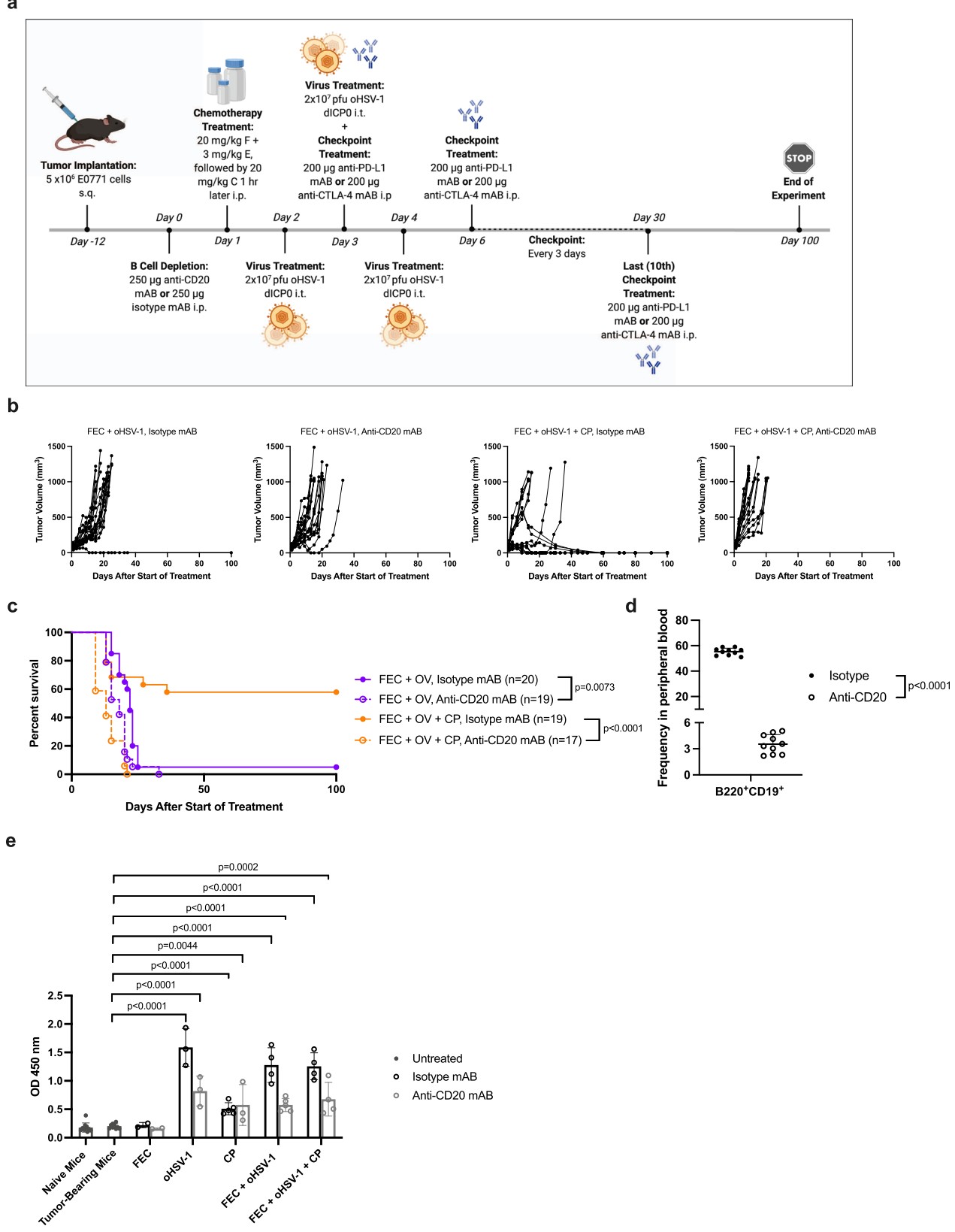

by day 15. The frequency of MDSCs in the blood (Fig. 7e) confirms that there is rapid expansion of the granulocytic MDSC population, with the frequency more than doubling in B-cell-depleted mice treated with triple therapy. Levels of other immune cell populations in the blood (CD4s, CD8s, monocytes, and DCs)

are consistent with expected findings, with no clear disturbance in their frequencies in the absence of B cells (Supplementary Fig. 8).

Further highlighting the clinical impact of these findings, we assessed the prognostic significance of the MDSC gene signature downregulated by FEC + oHSV-1 therapy

**Fig. 6 Depletion of B cells results in loss of efficacy in mice treated with FEC + oHSV-1. a** C57/Bl6 mice bearing E0771 tumors were treated with either an anti-CD20 antibody or isotype antibody, followed by treatment with saline, FEC, oHSV-1, CP, FEC + oHSV-1, or FEC + oHSV-1 + CP therapy. *Created using BioRender.com. **b** Tumor volumes were measured every 2–3 days from the start of treatment until mice reached endpoint. Each line represents individual mice within the group. **c** Kaplan–Meier survival curves of combination therapy treatments, with both the depletion and isotype antibodies. **d** Flow cytometry confirmed that a singular dose of an anti-CD20 antibody is sufficient to deplete B cells within 24 h of administration. **e** Bar plot showing IgG production levels from naïve mice, tumor-bearing mice, and therapeutically treated mice. *Mantel–Cox test was used for statistical analyses in **c**, two-tailed paired *t* test was used for statistical analyses in **d**, and two-tailed unpaired *t* test was used for statistical analyses in **e**. Error bars are representative of the standard deviation.

**Table 1 MDSC genes differentially expressed between FEC + oHSV-1 and oHSV-1 treatments.**

| Gene | O vs. S | F + O vs. O | Function |
|---|---|---|---|
| S100A9 | 2730.36 | −9.79 | Associated with MDSC-mediated resistance to chemotherapy[67,68] Implicated in MDSC-driven metastasis[69] Associated with MDSC activation and suppressive function[70] |
| CXCL3 | 411.80 | −5.87 | MDSC activation and recruitment[71,72] |
| CXCR2 | 345.58 | −6.20 | Phenotypic marker of MDSCs[73] Myeloid cell differentiation and migration in the tumor microenvironment[72,74] MDSC recruitment[75] |
| CSF3 | 282.48 | −9.38 | Gene encoding G-CSF, critical to the accumulation of MDSCs[69] |
| S100A8 | 242.56 | −6.44 | Implicated in MDSC-driven metastasis[69] |
| CXCL2 | 233.83 | −5.11 | MDSC proliferation and chemotaxis[67,72] |
| CXCL1 | 120.11 | −7.78 | MDSC activation and recruitment[72] MDSC activation and recruitment[72] |
| TREM1 | 45.37 | −3.01 | MDSC marker[76] |
| NOS2 | 44.63 | −4.06 | MDSC-driven metastasis and T-cell suppression[69] |
| ARG2 | 41.34 | −3.11 | Associated with MDSC-mediated suppression[70] |
| LY6G | 16.52 | −7.04 | Phenotypic marker of MDSCs[72,73] MDSC-driven metastasis[69] |
| SLPI | 10.04 | −1.96 | MDSC differentiation[77,78] |
| FPR2 | 9.38 | −2.00 | Receptor for SAA3, a well-known inflammatory factor that connects MDSCs with cancer progression[79] |
| IL1R1 | 5.43 | −4.11 | MDSC-driven metastasis[80] |
| ADAM19 | 5.34 | −3.47 | STAT pathway differentiation of MDSCs[81] |
| SERPIND1 | 4.41 | −4.23 | MDSC proliferation and migration[82] |
| ITGB2 | 3.73 | −1.98 | MDSC effector function[83] |
| LY6C1 | 3.05 | −4.17 | Phenotypic marker of MDSCs[67,72,73,84] |
| PEAR1 | 2.76 | −3.47 | MDSC differentiation[85] |
| FYN | 2.02 | −1.71 | Differentiates MDSCs from DCs[86] |
| CD38 | 1.94 | −1.39 | Phenotypic marker of MDSCs[73] |
| SOCS1 | 1.65 | −1.25 | MDSC induction[87] |
| ATRAID | 1.22 | −1.21 | MDSC differentiation[73,88,89] |

(Table 1) in a combined cohort of publicly available clinical microarray data[34]. Mean expression of this signature above median was associated with worse overall survival (OS) in TNBC restricted patient data [HR = 2.08 (1.05–4.10); logrank *p* value = 0.031] (Fig. 7f), and generally across all breast cancer patients combined [HR = 1.73 (1.32–2.27); logrank *p* value < 0.01] (Fig. 7g). This suggests that attenuation of the MDSC signature by FEC + oHSV-1 could have positive prognostic implications.

To rule out the clonal effect and assess whether or not the correlation between B cells and MDSCs holds true across more than one TNBC tumor model, we also conducted immune analysis using the PY230 murine breast cancer line (Supplementary Fig. 9). Here too, we see that when mice are treated with FEC + oHSV-1 + CP and depleted of circulating B cells, there is a prominent expansion of MDSCs in the blood. These data suggest that the correlation between B cells and MDSCs is not due to tumor clonality, but rather carries over between different models.

**B cells are required to alleviate tumor immunosuppressive mechanisms**. To assess whether changes seen in the peripheral blood were consistent and representative of what was happening in the tumor, immune analysis was performed on TILs. E0771 tumors were grown in C57/Bl6 mice and treated with saline, FEC, oHSV-1, CP, FEC + oHSV-1, or FEC + oHSV-1 + CP in both the presence (isotype mAb) and absence (anti-CD20 mAb) of circulating B cells. Mice were sacrificed on day 10 and TILs were processed and stained for analysis via flow cytometry. Consistent with previous findings, treatment with FEC + oHSV-1 or FEC + oHSV-1 + CP increases the level of TIL-Bs (Fig. 8a), particularly those of a memory phenotype (CD19$^+$CD27$^+$). Interestingly, whether or not mice were depleted of circulating B cells did not significantly affect the level of TIL-Bs. However, in line with findings from the peripheral blood, depletion of circulating B cells did result in a rapid expansion of MDSCs (Fig. 8b, c). Irrespective of whether mice were treated with the depletion antibody or not, the triple combination therapy was also shown to significantly reduce immunosuppressive populations in the tumor, such as the CD244.2$^+$ immunoregulatory receptor, PDL1$^+$ immune checkpoint, and F4/80$^+$ tumor-associated macrophages. The relationship between B cells and myeloid cells was further shown in both splenocytes (Fig. 8e, f) and tumor-draining lymph nodes (TDLNs) (Fig. 8g–i). The levels of other immune cell populations in the tumor, spleen, and TDLN (CD4s, CD8s, and DCs) are consistent with expected findings, with

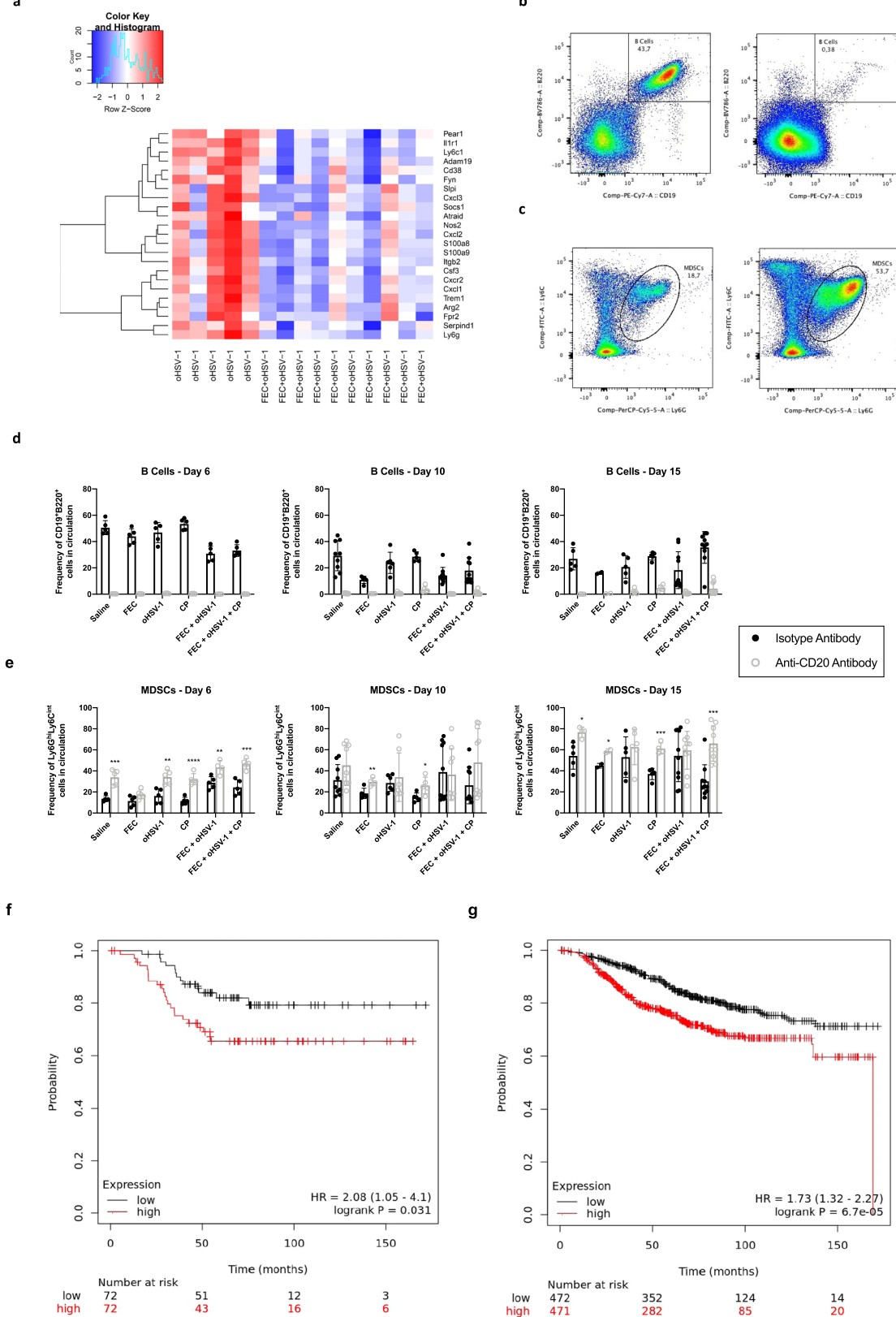

no clear disturbance in their frequencies in the absence of B cells (Supplementary Fig. 10).

**FEC + oHSV-1 responders present with a B-cell gene signature and control of MDSCs.** Consistent with clinical outcomes, mice treated with FEC + oHSV-1 have dichotomous responses to therapy, shown by the survival outcomes (Fig. 1), IHC quantification (Fig. 4), and gene expression profiles (Fig. 5). To further identify genomic features influencing outcome to therapy, we have sub-grouped these mice into what we believe to be responders and non-responders to treatment, as based on their

**Fig. 7 Absence of circulating B cells results in rapid expansion of granulocytic MDSCs.** C57/Bl6 mice bearing E0771 tumors were treated with either an anti-CD20 mAB or isotype mAB, followed by treatment with saline, FEC, oHSV-1, CP, FEC + oHSV-1, or FEC + oHSV-1 + CP. Blood was drawn on days 6, 10, and 15 and analyzed via flow cytometry. **a** Heat map showing selected MDSC-related genes and their expression across all oHSV-1 and FEC + oHSV-1 samples, as determined by whole tumor RNA sequencing analysis. **b** Representative flow plot showing the gating strategy for B cells (CD19+B220+ cells) in mice treated with the isotype mAB (left) and an anti-CD20 mAB (right). **c** Representative flow plot showing the gating strategy for MDSCs (Ly6G$^{hi}$Ly6C$^{int}$ cells) in mice treated with the isotype mAB (left) and the anti-CD20 mAB (right). **d** Bar plots showing the frequency of B cells in circulation across all timepoints. **e** Bar plots showing the frequency of MDSCs in circulation across all timepoints. **f** Kaplan–Meier survival plot for TNBC patients OS ($n = 144$) based on mean expression levels of a signature comprised of all available genes from Table 1. High versus low expression defined as above or below median expression. Logrank $p$ value and hazard ratio (95% confidence interval) are displayed. **g** Kaplan–Meier survival plot for breast cancer patients OS ($n = 943$) based on mean expression levels of a signature comprised of all available genes from Table 1. High versus low expression defined as above or below median expression. Logrank $p$ value and hazard ratio (95% confidence interval) are displayed. *Two-tailed unpaired $t$ test was used for statistical analyses. Error bars are representative of standard deviation.

likeness to the gene expression profiles of saline-treated mice (Fig. 9a). Assessment of the differentially expressed gene list shows that responders have upregulation of many genes associated with B-cell receptor signaling pathways (Fig. 9b), as consistent with previous findings. Additional to this and perhaps even more notable, the responders also downregulate *Siglec15*, a critical immune suppressor that is commonly upregulated on human cancer cells and tumor-infiltrating myeloid cells[39].

We utilized a publicly available cohort of combined clinical microarray data containing response rates to various therapies[40]. While there are no large datasets available for breast cancer patients treated with oncolytic virotherapy, evaluation of CD19 mRNA expression in patients treated with FEC therapy indicated CD19 trended towards being a predictor of complete pathological response in a Her2(-) ER(-) restricted patient group [AUC = 0.657, ROC $p$ value = 0.02, Mann–Whitney $p$ value = 0.054], and was a predictor of pathological response across combined breast cancer patients treated with FEC [AUC = 0.729, ROC $p$ value < 0.01, Mann–Whitney $p$ value < 0.01].

## Discussion

Immune checkpoint blockade has surged to the forefront of cancer therapy with astonishing clinical success rates and low toxicity profiles. However, this highly efficacious therapy only works in a fraction of patients and we have yet to fully elucidate the underlying biology that allows some patients to respond to therapy, while others do not. Helmink and colleagues have eloquently shown that B cells play an important role in promoting efficacious responses to immune checkpoint blockade in melanoma and renal cell carcinoma patients[41]. In line with these findings, our data suggest that promoting a B-cell signature within the tumor allows for successful treatment with combination immunotherapy platforms. Using the clinical chemotherapy cocktail FEC, in combination with oHSV-1, we were able to elicit the upregulation of B-cell receptor signaling pathways that allowed our otherwise non-responsive TNBC tumors to respond to CP. In particular, mice treated with the triple combination therapy had a re-population of circulating B cells over time and an increase in memory B cells within the tumor, suggesting that the sustained presence of B cells may be required to achieve durable responses and improved prognostic outcomes.

In vivo depletion studies in which an anti-CD20 antibody was used to deplete mice of circulating B cells have shown that groups of mice that would otherwise achieve complete responses to treatment instead had a complete loss of therapeutic efficacy. Deeper investigation into this phenomenon revealed a strong correlation between the presence of B cells and control of immunosuppressive MDSCs. These data suggest that B cells are required to suppress the rapid expansion of immunosuppressive myeloid cell populations in the TME.

Indeed, RNA sequencing data identified that FEC + oHSV-1 is able to not only upregulate genes associated with the B-cell lineage, but also to downregulate genes that are known to be key players in tumor immunosuppression. When further looking at the population of mice with the most distinct RNA profile (as thought to be responders to treatment) we find that amongst the downregulated MDSC genes is *Siglec15*, a known immune suppressor broadly expressed on human cancer cells and tumor infiltrating myeloid cells[39]. As Chen and colleagues have shown, Siglec15 overexpression has been documented in numerous human cancers and its expression is mutually exclusive of PD-L1, suggesting that it may be a potential therapeutic target for patients who are refractive to anti-PD-1/PD-L1 checkpoint blockade therapy[42]. Interestingly, our therapy downregulates this key immune modulator, suggesting a therapeutic platform that can be used to target this pathway of tumor immune escape. Along with downregulation of Siglec15, immune analysis studies showed that our triple combination therapy significantly reduces levels of CD244.2, an immunoregulatory receptor found on a variety of immune cells, including exhausted CD8+ T cells and MDSCs[43].

Current immunotherapies do not target or consider B cells, despite their predominance in the TME and key role in the adaptive immune response. In TNBC, evidence suggests that TIL-Bs generate a robust humoral response to amplify antitumor immunity[44] and mediate immunotherapy outcomes[45]. TIL-Bs have been correlated with enhanced overall survival[46], but findings in the literature suggest conflicting roles with TIL-Bs having both pro- and anti-tumorigenic functions. TIL-Bs have been identified as mediators of malignancy in several cancer types, with TIL-B depletion yielding positive outcomes[47]. Conversely, coordinated antibody and T-cell responses have been documented in cancer patients and TIL-Bs have been correlated with improved outcome[41,45,48–50]. Further, clinical studies have reported that TIL-Bs are influenced by the TME and immunotherapy, a discovery consistent with our findings. Indeed, our B-cell gene signature correlates with improved RFS in TNBC patients as well as all breast cancer subtypes combined, highlighting the clinical impact of these findings.

MDSCs are a heterogenous population of immature myeloid cells that accumulate in tumor beds and lymphoid organs (such as the spleen) of tumor-bearing hosts. MDSCs have long been attributed to tumor immunosuppression, most commonly due to their ability to suppress T-cell-mediated immune responses. However, studies have also shown that granulocytic MDSCs can suppress antitumor B-cell responses, predominantly through the secretion of nitric oxide (NO), arginase (Arg), and IL1[51–53]. While there are no documented studies to our knowledge looking at the reverse regulation of this phenomenon, our data suggests that while MDSCs can suppress TIL-Bs through the

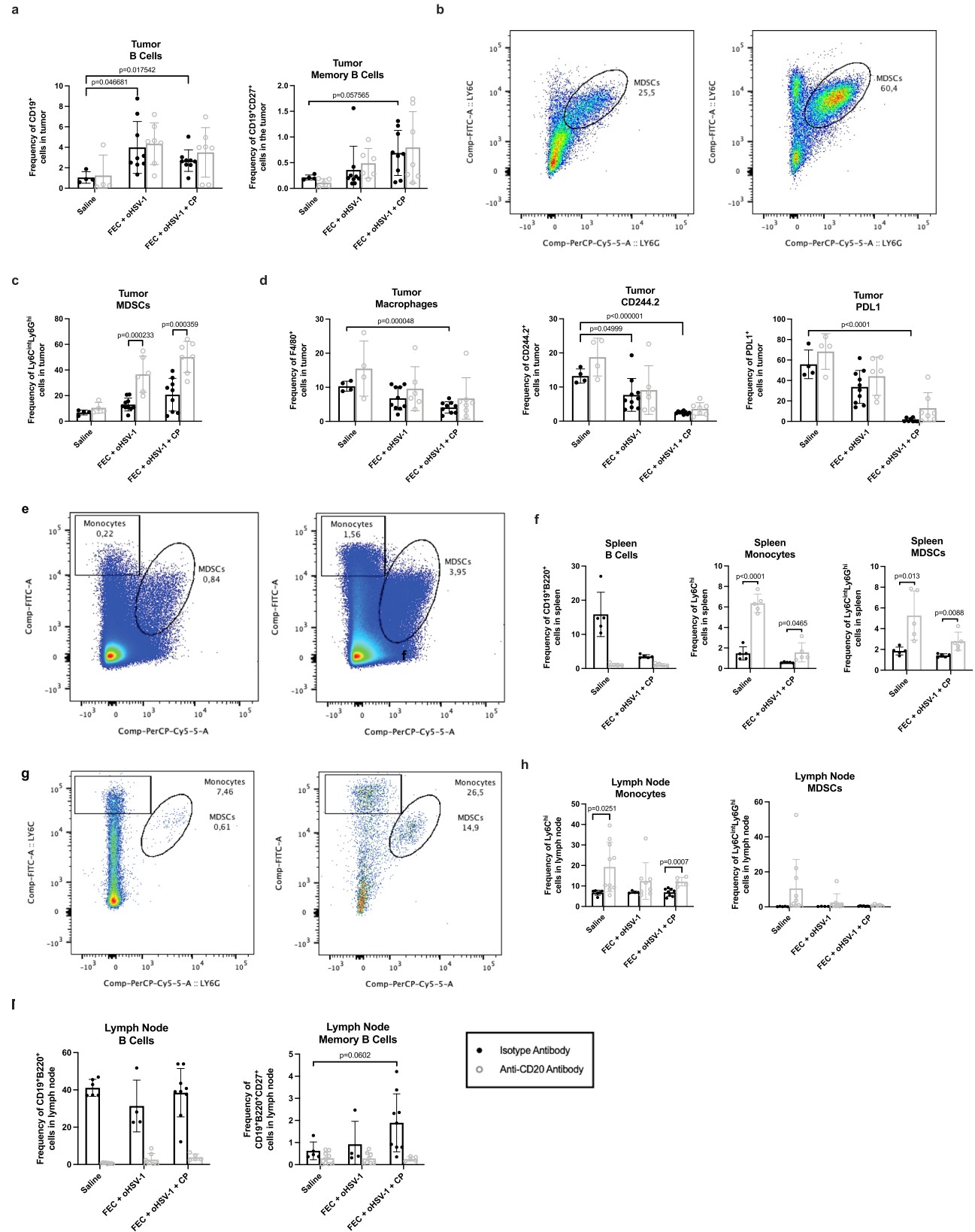

secretion of NO, Arg, and IL1 perhaps the opposite is also true. We have shown that our therapeutic platform upregulates B-cell receptor signaling pathways, suppressing key genes such as *NOS2*, *Arg2*, and *IL1R1* (Table 1), which we believe to be essential for the regulation and control of MDSCs. Indeed, the MDSC gene signature that we identified from our RNA sequencing data showed a strong correlation with clinical datasets, with low expression of those genes correlating with improved OS in TNBC patients as well as all breast cancer subtypes combined.

**Fig. 8 FEC + oHSV-1 + CP increases TIL-Bs and reduces immunosuppressive cell populations in E0771 tumors C57/Bl6 mice bearing E0771 tumors were treated with either an anti-CD20 mAB or isotype mAB, followed by treatment with saline, FEC + oHSV-1, or FEC + oHSV-1 + CP.** Mice were sacrificed on day 10, tumors were processed, and TILs were stained for analysis via flow cytometry. **a** Bar plots showing the frequency of B cells (CD19+) and memory B cells (CD19+CD27+) in the tumor. **b** Representative flow plot showing the gating strategy for MDSCs (Ly6GhiLy6Cint) in mice treated with the isotype mAB (left) and the anti-CD20 mAB (right). **c** Bar plots showing the frequency of MDSCs in the tumor. **d** Bar plots showing the frequency of macrophages (F4/80+), CD244.2+ cells, and PDL1+ cells. **e** Representative flow plot showing the gating strategy for monocytes (Ly6Chi) and MDSCs in splenocytes. **f** Bar plots showing the frequency of B cells (CD19+B220+), monocytes, and MDSCs in splenocytes. **g** Representative flow plot showing the gating strategy for MDSCs in TDLNs. **h** Bar plots showing the frequency of monocytes and MDSCs in TDLNs. **i** Bar plots showing the frequency of B cells and memory B cells in TDLNs. *Two-tail unpaired *t* test was used for statistical analyses. Error bars are representative of standard deviation.

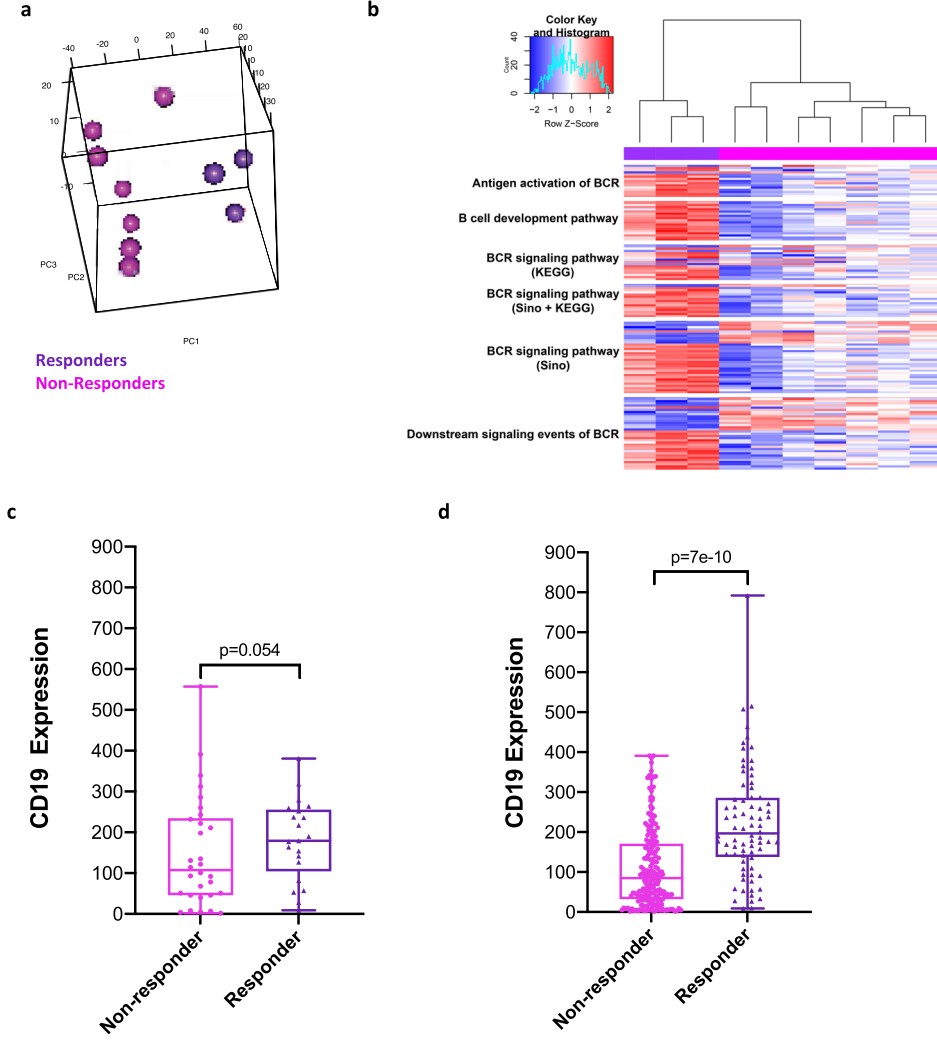

**Fig. 9 FEC + oHSV-1 induces dichotomous response, with responders having upregulation of B-cell receptor signaling pathways and downregulation of genes associated with immunosuppressive phenotypes. a** 3-D cluster plot showing the RNA expression correlations between mice treated with FEC + oHSV-1 (non-responders = pink; responders = purple). **b** Heat map showing the normalized expression values of genes associated with B-cell receptor signaling pathways, across all FEC + oHSV-1 samples. **c** CD19 mRNA expression was compared between responders (*n* = 23) versus non-responders (*n* = 30) to FEC treatment in a cohort of HER2(-) ER(-) breast cancer patients [AUC = 0.657 Mann–Whitney *p* value = 0.054]. **d** CD19 mRNA expression was compared between responders (*n* = 84) versus non-responders (*n* = 219) to FEC treatment in a cohort of all breast cancer patients [AUC = 0.729, Mann–Whitney *p* value < 0.001]. Response defined as complete pathological response versus residual disease after completing therapy. *BCR B Cell Receptor.

We have used a multi-pronged therapeutic approach to treat triple-negative breast tumors (Fig. 10). In this approach, we have combined low-dose chemotherapies that work to enhance tumor immunogenicity and antigen presentation aiding in BCR-recognition of TAAs. This, coupled with our oncolytic virotherapy initiates a cascade of DC and B-cell recruitment and ultimately clonal expansion of B cells. B-cell and T-cell priming

results in the release of antitumor chemokines driving effector cell function, while simultaneously suppressing cytokines responsible for the accumulation of MDSCs in the TME. This immune-activating cascade of events leads to an immune landscape conducive to successful treatment with CP.

Our studies are limited by the nature of murine hosts and their inability to fully recapitulate human biology. It is common to

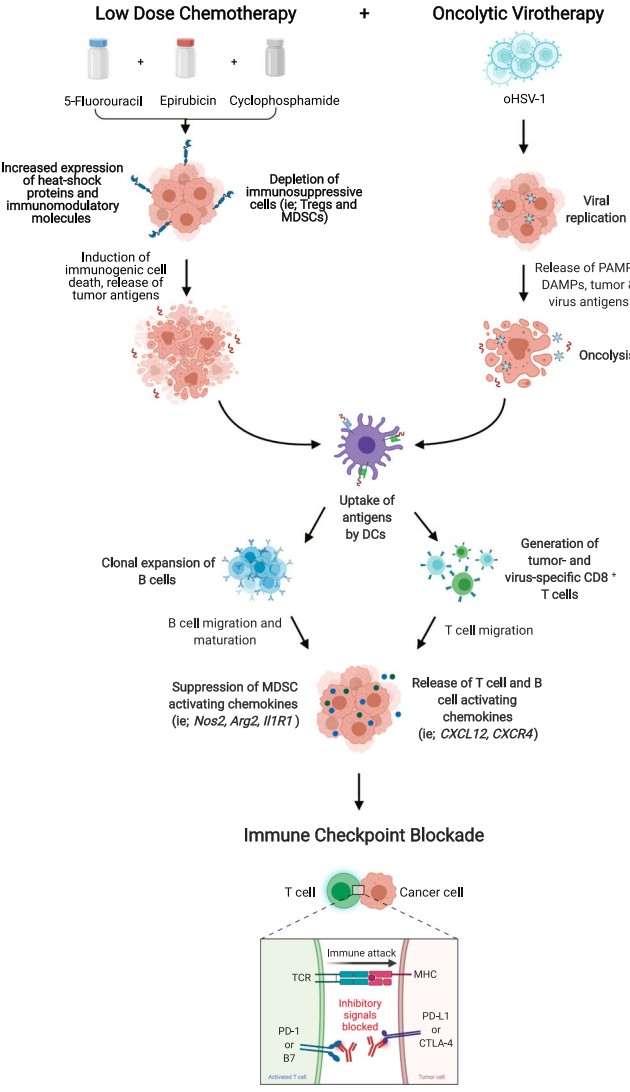

**Fig. 10 Schematic of therapy-induced immune activation.** *Created using BioRender.com.

have patients with phenotypically similar cancers present with varying response levels to treatment. This phenomenon was also seen in our studies, in which mice with identical genetic makeup had dichotomous responses to the same treatment. This inter-mouse heterogeneity may be due to the individual metabolic activity and immune profile of each individual mouse. Our data were generated from subcutaneous tumors and should be further analyzed in metastatic and spontaneously arising tumor models to assess these core immunological interactions in a more naturally occurring setting. We aim to further assess these fascinating findings as we continue to phenotype our B-cell populations and assess their interactions with various myeloid cell populations in cancer-bearing hosts. We believe that clinical studies should focus not only on the levels of T cells and their correlated effects to prognostic outcomes, but also look at B cells as a key biomarker to predict responses to immunotherapy treatments.

## Methods

**Cell lines**. Human osteosarcoma cells from a fifteen-year old Caucasian female (U2OS; ATCC, Manassas, VA) were maintained in Dulbecco's modified Eagle's media (DMEM) supplemented with 10% fetal bovine serum (FBS, ATCC 30-2020) 2 mmol/l L-glutamine, 100 U/ml penicillin, and 100 μg/ml streptomycin (Gibco, Grand Island, NY). Vero cells originating from the kidney of an adult monkey

(ATCC, Manassas, VA) were maintained in DMEM supplemented with 10% FBS 2 mmol/l L-glutamine, 100 U/mL penicillin, 100 μg/mL streptomycin (Gibco, Grand Island, NY). Murine medullary breast adenocarcinoma cells isolated as a spontaneous tumor from a C57/Bl6 mouse (E0771; CH3 Biosystems, Amherst, NY) were maintained in roswell park memorial institute (RPMI) medium supplemented with 10% FBS, 10 mM HEPES, 100 U/mL penicillin, 100 μg/mL streptomycin, and 2 mmol/l L-glutamine. Murine breast adenocarcinoma cells derived from an MMTV-PyMT tumor were maintained in Ham's F-12K medium supplemented with 5% FBS and 0.1% MITO + serum extender (Corning #355006). All cell lines were grown at 37 °C with 5% $CO_2$.

**Mouse experiments**. Mice were maintained at the McMaster University Central Animal Facility and all the procedures were performed in full compliance with the Canadian Council on Animal Care and approved by the Animal Research Ethics Board of McMaster University. Six- to eight-week-old female C57/Bl6 mice (Charles River Laboratories, Wilmington, MA) were used to implant $5 \times 10^6$ E0771 cells subcutaneously on the left flank. Mice were weighed and all found to be approximately 20 g in size. Mice were housed in groups, 5/cage, fed a normal diet, and kept at room temperature. To minimize experimental variability, low passage E0771 cells were used for subcutaneous injections. Twelve days after injection, the tumors reached treatable average tumor volume (50–100 mm³). Mice were blindly randomized prior to the start of treatment. In experimental groups receiving FEC treatment, mice were treated on day 1 with 20 mg/kg 5-fluorouracil in 200 μL saline, followed by 3 mg/kg epirubicin in 200 μL saline, followed by 20 mg/kg cyclophosphamide in 200 μL saline 1 h later. In experimental groups receiving AC treatment, mice were treated on day 1 with 3 mg/kg doxorubicin in 200 μL saline, followed by 20 mg/kg cyclophosphamide in 200 μL saline 1 h later. All chemotherapy injections were given intraperitoneally (i.p.). Experimental groups receiving oHSV-1 were treated with $2 \times 10^7$ pfu oHSV-1 dICP0 in 50 μL PBS intratumorally (i.t.) on days 2, 3, and 4. Experimental groups receiving immune checkpoint blockade therapy were treated with α-CTLA-4 (BioXCell, BE0131) and α-PD-L1 (BioXCell, BE101) antibodies (200 μg/200 μL PBS, each) starting on day 3, every 3 days until mice reached endpoint or a total of 10 doses had been given. For B-cell depletion studies, mice were treated on day 0 with a singular dose of 250 μg of α-CD-20 (Biolegend, 152104) or isotype (Biolegend, 400566) antibody. For all mouse studies, tumors were measured every 2–3 days and mice having a tumor volume of 1000 mm³ were classified as end point.

**Drug preparation**. 5-fluorouracil stock powder (Sigma Aldrich, F6627) was stored at 4 °C and dissolved in sterile saline to a concentration of 2 mg/mL. Epirubicin stock powder (Cayman Chemicals, 12091) was stored at −20 °C and dissolved in sterile saline to a concentration of 0.3 mg/mL. Cyclophosphamide stock powder (Sigma Aldrich, C0768) was stored at 4 °C and dissolved in sterile saline to a concentration of 2 mg/mL. Doxorubicin stock powder (Sigma Aldrich, D1515) was stored at 4 °C and dissolved in sterile saline to a concentration of 0.3 mg/mL. α-CTLA-4 (BioXCell, BE0131) and α-PD-L1 (BioXCell, BE0101) antibodies solutions were diluted to 1 mg/mL with sterile PBS. All solutions were prepared fresh for each experiment.

**Virus preparation**. Recombinant HSV-1 was generated by homologous recombination using infectious DNA of luciferase-expressing wild-type HSV-1 KOS/Dluc/oriL[54]. HSV-1 dNLS encodes a GFP-tagged protein that lacks the ICP0 NLS region and a portion of the C-terminal oligomerization domain[55]. HSV-1 dICP0 contains a deletion of the entire ICP0 coding region. All HSV-1 ICP0 mutants were propagated and tittered on U2OS cells in the presence of 3 mmol/l hexamethylene bisacetamide (Sigma, St Louis, MO). Wild-type HSV-1 strain KOS was propagated and titered on Vero cells. All viruses were purified and concentrated via sucrose cushion ultracentrifugation and purified virus was resuspended in PBS and stored at −80 °C.

**Rechallenge experiment**. Mice that achieved a complete response to therapy (tumor-free mice) and naïve mice, as control, were subcutaneously implanted with $5 \times 10^6$ E0771 cells in the left flank. Tumors were measured every 3–4 days for a minimum of 4 weeks.

**Cytokines analysis**. Mice were anaesthetized and euthanized before resection of the tumors. As described[56], tumors were cut into small pieces and homogenized in the presence of tissue extraction solution (50 mM Tris, pH 7.4, 250 mM NaCl, 5 mM EDTA, 2 mM Na3VO4, 1 mM NaF, 20 mM Na4P2O7, 1 mM beta-glycer-ophosphate, 1% NP-40). Homogenized tumors were incubated on ice for 30 min. Whole-tumor lysates were clarified by three sequential centrifugations at 14,000 rpm for 10 min at 4 °C. Tumor homogenates were diluted to achieve equal amounts of protein concentration. Forty-four-Plex murine cytokine/chemokine analysis was done by Eve Technologies (Calgary, Alberta, Canada).

**Histology and image analysis**. Treated and control tumors were resected on day 7, fixed in 10% formalin for 48 h, and then transferred to 70% ethanol until histological processing. Tumor tissue was embedded in paraffin and 4-μm sections

were prepared. Tissue sections were processed for hematoxylin staining and IHC using Automated Leica Bond Rx stainer with Epitope Retrieval Buffer 2 (Leica, AR9640). All antibodies were diluted in IHC/ISH Super Blocker (Leica, PV6199). Primary antibodies and working dilutions were as follows: α-CD3 (1:150; Abcam, ab16669), α-CD4 (1:800; eBio, 14-9766), α-CD8a (1:1000; eBio, 14-0808), α-FOXP3 (1:100; eBio, 14-5773-82), and α-Ly6G (1:1000; Biolegend, 127602). Secondary antibody and working dilution: rabbit α-rat antibody (1:100; Vector Labs, BA-4001). Bond Refine Polymer Detection kit (Leica, DS9800) was used. Additional tissue sections were processed for immunofluorescence (IF) staining. Slides were incubated at 37 °C overnight, deparaffinized, and rinsed with distilled water (dH$_2$O). Slides were fixed in 10% neutral buffered formalin (NBF) for 20 min and rinsed with dH$_2$O. Antigen retrieval was performed using the decloaking chamber plus with Diva decloaker (Biocare). Slides were added to the Intellipath FLX rack, rinsed with dH$_2$O, and covered with TBS buffer. Slides were loaded into the pre-programmed, reagent-loaded Intellipath FLX. Endogenous peroxidase blocking was performed by adding peroxidazed-1 for 5 min at room temperature (RT), followed by nonspecific blocking with Rodent Block M for 30 min at RT. Primary antibody staining was performed in five rounds: CD3 (1:600, Spring Bioscience Corp., M3074), PNAd (1:200, Biolegend, 120802), Pax5 (1:1000, Abcam, EPR3730), CD8 (1:200, Cell Signaling Technologies, 98941 S), CD11b (1:7500, Abcam, EPR1344). Each round was followed by application of Mach2 Rb HRP for 10 min at RT and fluorophore incubation for 10 min at RT. Diluted IF counterstain DAPI was applied for 5 min at RT and slides were rinsed with water. Finally, Vectashield mounting media and cover slip were applied and slides were stored in the dark at RT. All images were scanned with Vectra3 and annotated using Phenochart software.

**RNA sequencing**. Mice were sacrificed on day 5 and tumors harvested for RNA sequencing analysis. Tumors were homogenized in 1 mL trizol (Thermo Fisher Scientific, #15-596-018). A total of 375 μL was transferred to 1250 μL trizol and mixed well. RNA was first extracted using chloroform and then following the manufacturer's instructions using an RNeasy RNA extraction kit (Qiagen). cDNA libraries were created by polyA enrichment using NEBNext poly(A) magnetics isolation module (NEB) and reverse transcribed using NEBNext ultra II directional RNA library prep kit (NEB) according to the manufacturer's instructions. cDNA libraries were sequenced using an Illumina HiSeq rapid V2 (1 ×50 bp sequence reads) at the Farncombe Metagenomics Facility (McMaster University). The sequencing was run in 2 batches. Sequencing yielded ~15 × 10$^6$ reads/sample. First, reads were filtered by quality (at least 90% of the bases must have a quality score of 20 and higher). Then the mapping of the remaining reads was performed using HISAT2[57] with hg38 (UCSC) reference genome; reads were counted by using HTSeq count[58]. Genes which did not show sufficiently large counts were removed using filterByExpr function in EdgeR package[59,60] in R, resulting in 13,079 and 12,334 genes in the first and second batches, respectively. These remaining count values were normalized with TMM normalization method[61] and then transformed with voom transformation[62]. 12,334 genes were shared between the first and second batches and were used for batch effect removal using ComBat[63], with experiment date used as the batch information. Limma package[64] in R was used to examine differential expression between the groups of interest; p-values obtained from the analysis were corrected with BH correction for multiple testing[65], and corrected values <0.05 were considered to be significant.

**ELISA assay**. Cell lysates of E0771 tumor cells were prepared by sonication. Polystyrene 96-well microtiter plates (Nunc MaxiSorp; Thermo Fisher Scientific) were coated overnight at 4 °C with 100 μL of E01771 tumor cell lysate (5 μg/mL in PBS). The wells were washed 3x with wash buffer (0.05% Tween in PBS) and blocked with 300 μL of blocking buffer (1% bovine serum albumin (BSA) in PBS) for 3 h at RT. The wells were washed 3x with wash buffer. 50 μL of a 1/50 dilution of each test or control mouse serum, diluted in 1% BSA-PBS, was added in triplicate and incubated for 3 h at RT. The wells were washed 3x with wash buffer, and 100 μL of horseradish peroxidase conjugated goat anti-mouse IgG Fc antibody (1:10,000; Thermofisher Scientific, cat# A16084) diluted in blocking buffer was added for 1 h at RT. Wells were washed 3x with wash buffer and 3,3′,5,5′-Tetramethylbenzidine (TMB; BD Biosciences) substrate solution was added for 20 min at RT. Stop solution (2 N H$_2$SO$_4$) was added to each well to stop the reaction and the 450 nm and 540 nm OD was read using a SpectraMax i3 Microplate Autoreader (Molecular Devices).

**Survival analysis: Kaplan–Meier and ROC plots**. Kaplan–Meier survival plots (Figs. 5e, f and 7f, g) were generated using the publicly available online software housed on KMplot.com[34]. This online database comprises numerous Breast Cancer microarray datasets derived from publicly available datasets available on NCBI Gene Expression Omnibus (GEO). The combination and normalization of qualifying datasets for analysis through this tool is described by the developers in Győrffy et al.[34]. Patients with clinical annotation were analyzed broadly, or after discrimination for patients fitting TNBC characteristics according to IHC-negative status of ER, PR, and array-based negative status of HER2. Probes were selected based on JetSet optimization. For signature analysis, the mean expression of the top available 29 genes (B cell Signature) or all available (19) genes (MDSC Signature)

was analyzed according to mRNA expression above (high) or below (low) either median or upper tertile. The resulting hazard ratio and Logrank p value are displayed for RFS or OS. Prognostic significance of CD19 (Fig. 9c, d) was assessed using the publicly available online software housed on ROCplot.org[40]. Patient cohort microarrays with treatment and response annotation were accrued by the developers utilizing NCBI Gene Expression Omnibus (GEO)[40]. Pathological complete response was defined as pathological complete response versus residual disease after completing therapy. CD19 mRNA expression was assessed as a predictor of pathological complete response in any patients treated with combined fluorouracil, epirubicin, cyclophosphamide (FEC). mRNA expression of CD19 was plotted for responders versus non-responders using Graphpad Prism. Outliers were included in statistical analysis but removed from graph for visualization purposes using the ROUT method with Q = 0.2%. Area under the curve (AUC) and Mann–Whitney p-value are disclosed. It is important to note when interpreting these analyses that the combined datasets used for these analyses may not reflect the observed clinical distribution of breast cancer patients and subtypes in a true population, because the patient population of each microarray was curated according to the purpose of that corresponding study. Nonetheless, they provide a large cohort of breast cancer expression data with clinical annotation, survival data, and therapeutic response data.

**Flow cytometry analysis**. Organs were harvested from animals at given time points. Tumors were minced with a razor blade in RPMI media. 100 μL liberase (Sigma Aldrich, #5401054001) was added for digestion and samples incubated for 20 min at 37 °C. The cell suspension was passed over a 100 micron filter and rinsed with 5 mL of RPMI. Samples were spun at 1500 RPM for 5 min. Spleens and lymph nodes were pressed between two glass slides to extract cells and 150 μL of blood was collected from the periorbital sinus. Red blood cells from all samples were lysed using ACK buffer. The PBMCs were treated with anti-CD16/CD32 (Fc block; BD Biosciences, #553141) and surface stained with fluorescently conjugated antibodies for FVS (BD Biosciences, #564406), CD19 (Fisher Scientific, #14-019-482), B220 (BD Biosciences, #563894), CD27 (BD Biosciences, #558754), CD4 (BD Biosciences, #561830), CD8 (BD Biosciences, #563046), CD11b (BD Biosciences, #553311), Ly6C (BD Biosciences, #553104), Ly6G (BD Biosciences, #560602), CD11c (BD Biosciences, #562782), F4/80 (BD Biosciences, #743282), CD244.2 (BD Biosciences, #740860), and PD-L1 (BD Biosciences, #563369). LSRFortessa flow cytometer with FACSDiva software (BD Biosciences) was used for data acquisition and FlowJo Mac, version 10.0 software was used for data analysis.

**Quantification and statistical analysis**. IHC slides were digitalized using the Olympus VS120-L100-W automated slide scanner. They were batch-scanned on the brightfield setting at 20x magnification. The color camera used was the Pike 505 C VC50. HALO Image Analysis Software (Indica Labs, HALO v2.2) was used to analyze digital histology image. Cytonuclear cell count algorithms were developed to determine the amount of CD3, CD4, CD8a, FOXP3, and Ly6G positive cells and total cell number present in a given sample of tissue. Percentage of positive cells was calculated relative to total cell number[66]. For each statistical analysis used, normality of the distributions and variance assumptions were tested before running the statistical analyses. Multiple t-tests were used to determine the statistical significance of the differences in means. The log-rank (Mantel–Cox) test was used to determine statistical significance for the difference in Kaplan–Meier survival curves between treatments. All the tests were two-sided. The null hypothesis was rejected for p-values less than 0.05. All data analyses were carried out using GraphPad Prism (La Jolla, CA, USA).

**Statistics and reproducibility**. E0771 tumors take approximately 12 days to reach treatable size (50–150 mm$^3$). Experiments were conducted with 5–10 mice per group. In vivo studies were conducted in double replicates where possible. Cytokines analysis was conducted with triple replicates.

For each statistical analysis used, normality of the distributions and equality of variance assumptions were tested before running the statistical analyses. One-way analysis of variance, non-parametric Kruskal–Wallis test, and t test were used to determine the statistical significance of the differences in means. For analyzing the statistical significance of the difference in Kaplan–Meier survival between treatments, the Log-rank (Mantel–Cox) test was used. All the tests were two-sided. The null hypothesis was rejected for p values <0.05. All data analyses were carried out using GraphPad Prism (La Jolla, CA, USA).

**Reporting summary**. Further information on research design is available in the Nature Research Reporting Summary linked to this article.

## Data availability

The gene list used to generate Fig. 5d can be found in Supplementary Data 1. The antibodies used can be found in Supplementary Data 2. The source data behind the graphs in this paper are available in Supplementary Data 3. Complete RNA sequencing data set can be found in the GEO database, GSE152698. All other data are available from the authors upon reasonable request.

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

## Acknowledgements

Alyssa Vito was the recipient of the Vanier Canada Graduate Scholarship. We would like to acknowledge Spencer Revill for his work on the IHC images and quantification. We thank Mary Smith and Mary Bruni of John Mayberry Core Histology Facility, McMaster University for immunohistochemistry staining. This work was sponsored by operating grants from the Canadian Cancer Society Research Institute (formerly the Canadian Breast Cancer Research Alliance); grant #319377 and #706280.

## Author contributions

Conceptualization, A.V. and K.L.M.; Methodology, A.V., O.S., K.M., A.A.A., S.T.W., and K.L.M.; Investigation, A.V., O.S., N.E., I.P.M., A.L.P., and D.H; Writing—Original Draft, A.V.; Writing—Review & Editing, A.V., O.S., N.E., I.P.M., A.L.P., K.M., D.H., A.A.A., Y.W., S.T.W., B.H.N., T.C.B., and K.L.M.; Funding Acquisition, A.V., S.T.W., and K.L.M.; Resources, Y.W., and K.L.M.; Supervision, K.M., A.A.A., Y.W., S.T.W., B.H.N., T.C.B., and K.L.M.

## Competing interests

The authors declare no competing interests.
