## [Peer Review File · Communications Biology]

Reviewers' Comments:

Reviewer #1:

Remarks to the Author:

It is very interesting research that explore a genuine idea about enhancing the effect of immunotherapy in Triple Negative Breast Cancer

In comparison to other solid tumor the utilization of check point inhibitors in Triple Negative Breast Cancer is still limited due to dismal response observed in phase III Clinical trials due to unique biology of triple negative breast cancer

In this study authors came with genuine hypothesis utilizing Oncolytic virus combined with low dose chemotherapy to increase tumor-infiltrating lymphocytes, in immune-bare tumors, which allowed achieveing complete responses .

The results are encouraging and can be translated to be utilized in human clinical trials

Reviewer #3:

Remarks to the Author:

Vito and colleagues have investigated the efficacy and molecular effects of combined FEC + oncolytic virotherapy in a genetically engineered mouse model of TNBC. Combination therapy increased TILs and resulted in regression in some subsets of mice. RNA expression profiling revealed that this combination elicited increased B cell and decreased MDSC gene expression. The therapeutic effect was found to require B cells. In the absence of B cells in this mouse model, MDSC expansion was observed. This study provides important and translationally impactful insights.

1. The major limitation of this study is that it focuses on a single mouse model. Can authors provide data demonstrating these findings in an additional syngeneic mouse model to assess the generalizability of these findings? Authors should mine breast cancer RNA sequencing datasets to assess expression of their responder vs non-responder gene expression profile.

2. Please remove the statement in the abstract 'allowing mice to achieve complete responses to CP.' This should be edited to more accurately reflect the result – that 10% of mice achieved durable regression. See comment below for how to more rigorously assess 'complete' responses.

Regarding the observed 'complete responses' / 'cures':

-Authors should remove the term 'complete response' and 'cure' from throughout the manuscript. Instead, please precisely state the observed result. Lack of primary tumor palpation is not evidence of a cure.

-to truly assess lack of systemic disease, CTC analyses and quantitative detection of multi-organ metastases and disseminated tumor cells would need to be undertaken

-At a minimum, authors need to assess lung metastatic burden in vehicle/singly/combo treated mice.

3. Authors should address why the combination with FEC causes regression in some mice, but the combination of oHSV with AC does not.

4. Can authors correlate baseline tumor characteristics with eventual combination therapy responses to more clearly understand why some mice respond and others do not? Is this related to perhaps stochastic levels of baseline immune population infiltration in these tumors?

a. Authors could assess immune subpopulation heterogeneity in a cohort of untreated mice to appreciate whether this heterogeneity could explain the rarity of observed regressions.

Reviewer #4:

Remarks to the Author:

Synopsis:

In this manuscript, Vito et al. investigated the effects of low dose chemotherapy combined with oncolytic virotherapy in an E0771 murine TNBC tumor model. They found that the combination therapy increased tumor-infiltrating lymphocytes, contributing to improved responses to checkpoint blockade. The authors demonstrated that depletion of circulating B cells by anti-CD20 antibodies resulted in the expansion of MDSC and loss of therapeutic efficacy, suggesting that circulating B cells can act as regulators of MDSCs, a key population of cells that drive immune escape and mediate therapeutic resistance.

Criticism:

I think there are some interesting findings in this study: 1) The combination therapy (FEC + oHSV-1) improves overall survival, which is associated with increased tumor-infiltrating lymphocytes. 2) Circulating B cells are required for therapeutic efficacy of the combination therapy. 3) Depletion of circulating B cells affects the immunosuppressive cell populations, including MDSC. Although the finding #1 has been proposed in many studies [e.g., reviewed in DOI: 10.2147/OV.S66083], the findings #2 and #3 may provide a better understanding of cancer immunotherapy for TNBC. However, as mentioned below, the current dataset appears limited in providing sufficient evidence to support the authors' conclusion. Listed below are the major points/flaws that preclude at this stage acceptance in Communications Biology.

Major Points:

1. One of the major concerns of this reviewer is that the immunostimulatory effects of the combination therapy (FEC + oHSV-1) could be potentially due to cancer cell death, not specific to the therapeutic strategy. For instance, is there any difference between the combination therapy and high-dose chemotherapy (that induces similar dose of cancer cell death) in their tumor model, in terms of their effects on cancer immunity? The authors need to confirm that the combination strategy is a key to induce immunogenic response in their TNBC model.
2. It is not clear whether the therapeutic effect of the combination therapy ("FEC + oHSV-1" and/or "FEC + oHSV-1 + CP") is mediated by the host immune system. The authors need to use immunocompromised mice to confirm some of their major findings demonstrated in Figures 1 and 2.
3. The molecular and/or cellular mechanisms of MDSC regulation by circulating B cell is not clear. Because the link between circulating B and MDSC is one of the main findings of this paper, further investigation seems to be necessary.
4. In Figure 4C, the authors showed a significant increase in CD4 T cell infiltration in "FEC + oHSV-1" group. However, in figure S9B, the frequency of CD4 T cells in tumors seems to be unaffected by "FEC + oHSV-1" treatment. Can the author explain this superficial discrepancy? If this can be due to quantification method (histology v.s. FACS), TIL analysis demonstrated in Figure 4 needs to be analyzed again by flow cytometer. Histological analysis is useful to get positional information, but less quantitative.

Minor Points:

1. Some of the statements are not supported by the demonstrated data. For example, the authors state that "While no monotherapy showed efficacy, the addition of oHSV-1 to the clinical FEC regimen (FEC + oHSV-1) delayed tumor growth in some mice (Figure 1B) and significantly extended survival, with a 10% overall cure rate (Figure 1C)". To support this statement, the authors need to compare the "FEC" v.s. "FEC + oHSV-1" group, but the p value for this comparison is not indicated in Figure 1C. It would be important to carefully reconsider each statement.

2. It would be important to show the specificity of anti-CD20 antibodies used in this study. For example, the authors may confirm that the populations of other immune cells (e.g., T cell, macrophage,,,) are not affected by anti-CD20 antibody treatment (at the early time point). Or, the authors may use other methods for B cell depletion.

Review #1 (Remarks to the Author):

It is very interesting research that explore a genuine idea about enhancing the effect of immunotherapy in Triple Negative Breast Cancer

In comparison to other solid tumor the utilization of check point inhibitors in Triple Negative Breast Cancer is still limited due to dismal response observed in phase III Clinical trials due to unique biology of triple negative breast cancer

In this study authors came with genuine hypothesis utilizing Oncolytic virus combined with low dose chemotherapy to increase tumor-infiltrating lymphocytes, in immune-bare tumors, which allowed achieving complete responses.

The results are encouraging and can be translated to be utilized in human clinical trials

Author Response to Reviewer #1:

The authors would like to thank Reviewer #1 for their time spent reading and analyzing our work. We appreciate their positive feedback and attention to the translational implications of our findings.

Reviewer #2 (Remarks to the Author):

Vito and colleagues have investigated the efficacy and molecular effects of combined FEC + oncolytic virotherapy in a genetically engineered mouse model of TNBC. Combination therapy increased TILs and resulted in regression in some subsets of mice. RNA expression profiling revealed that this combination elicited increased B cell and decreased MDSC gene expression. The therapeutic effect was found to require B cells. In the absence of B cells in this mouse model, MDSC expansion was observed. This study provides important and translationally impactful insights.

Author Response: The authors would like to thank Reviewer #2 for taking time to read through our manuscript, highlighting areas that need to be improved upon and also recognizing the translational impact of our findings.

1. The major limitation of this study is that it focuses on a single mouse model. Can authors provide data demonstrating these findings in an additional syngeneic mouse model to assess the generalizability of these findings? Authors should mine breast cancer RNA sequencing datasets to assess expression of their responder vs non-responder gene expression profile.

Author Response: In an effort to rule out the clonal effect, the authors have shown that this phenomenon between B cells and MDSCs also occurs in a second tumor model (PY230), as demonstrated in Figure S8 (Lines 474-486). Further, we agree with the reviewer's suggestion to mine breast cancer RNA sequencing datasets as this adds to the depth and translatability of our findings. We have added this analysis into the manuscript as Figures 5E, 5F, 7F, 7G, 9C, 9D (Lines 298-311, 442-461, 554-567), as shown below.

A

B

C

D

E

F

Fig. 5. FEC + oHSV-1 induces significant upregulation in RNA transcriptomes associated with immune processes and pathways and more specifically, B cell receptor signaling pathways. **(A)** 3-D cluster plot showing the RNA expression correlations between mice treated with saline (grey; n=5), FEC (pink; n=10), oHSV-1 (teal; n=5) and FEC + oHSV-1 (purple; n=10). **(B)** Bar plot illustrating the results of pathway enrichment analysis performed on samples from mice treated with FEC + oHSV-1, compared to those treated with saline alone. **(C)** Heat map showing the normalized expression values of genes across all samples. **(D)** Heat map showing the normalized expression values of genes associated with B cell receptor pathways. **(E)** Kaplan-Meier survival plot for TNBC patient RFS (n=220) microarray data based on mean mRNA expression level of the top 29 available genes in Table S1. **(F)** Kaplan-Meier survival plot for all breast cancer patients RFS (n=2032) based on mean mRNA expression levels of the top 29 available genes in Table S1. *BCR = B Cell Receptor.

Fig. 7. Absence of circulating B cells results in rapid expansion of granulocytic MDSCs.

C57/Bl6 mice bearing E0771 tumors were treated with either an anti-CD20 mAb or isotype mAb, followed by treatment with saline, FEC, oHSV-1, CP, FEC + oHSV-1 or FEC + oHSV-1 + CP. Blood was drawn on days 6, 10 and 15 and analyzed via flow cytometry. **(A)** Heat map showing selected MDSC-related genes and their expression across all oHSV-1 and FEC+ oHSV-1 samples, as determined by whole tumor RNA sequencing analysis. **(B)** Representative flow plot showing

the gating strategy for B cells (CD19⁺B220⁺ cells) in mice treated with the isotype mAB (left) and an anti-CD20 mAB (right). (C) Representative flow plot showing the gating strategy for MDSCs (Ly6G^{hi}Ly6C^{int} cells) in mice treated with the isotype mAB (left) and the anti-CD20 mAB (right). (D) Bar plots showing the frequency of B cells in circulation across all timepoints. (E) Bar plots showing the frequency of MDSCs in circulation across all timepoints. (F) Kaplan-Meier survival plot for TNBC patients OS (n=144) based on mean expression levels of a signature comprised of all available genes from Table 1. High versus Low expression defined as above or below median expression. Logrank p value and Hazard Ratio (95% confidence interval) are displayed. (G) Kaplan-Meier survival plot for breast cancer patients OS (n=943) based on mean expression levels of a signature comprised of all available genes from Table 1. High versus Low expression defined as above or below median expression. Logrank p value and Hazard Ratio (95% confidence interval) are displayed. *Two-tailed unpaired t test was used for statistical analyses. Error bars are representative of standard deviation.

Fig. 9. FEC + oHSV-1 induces dichotomous response, with responders having upregulation of B cell receptor signaling pathways and downregulation of genes associated with immunosuppressive phenotypes.

(A) 3-D cluster plot showing the RNA expression correlations between mice treated with FEC + oHSV-1 (non-responders = pink; responders = purple). (B) Heat map showing the normalized

expression values of genes associated with B cell receptor signaling pathways, across all FEC + oHSV-1 samples. (C) CD19 mRNA expression was compared between responders (n=23) versus non-responders (n=30) to FEC treatment in a cohort of HER2(-) ER(-) breast cancer patients [AUC=0.657 Mann-Whitney p value=0.054]. (D) CD19 mRNA expression was compared between responders (n=84) versus non-responders (n=219) to FEC treatment in a cohort of all breast cancer patients [AUC=0.729, Mann-Whitney p value<0.001]. Response defined as complete pathological response versus residual disease after completing therapy. *BCR = B Cell Receptor.

2. Please remove the statement in the abstract 'allowing mice to achieve complete responses to CP.' This should be edited to more accurately reflect the result – that 10% of mice achieved durable regression. See comment below for how to more rigorously assess 'complete' responses.

Regarding the observed 'complete responses' / 'cures'

-Authors should remove the term 'complete response' and 'cure' from throughout the manuscript. Instead, please precisely state the observed result. Lack of primary tumor palpation is not evidence of a cure.

-to truly assess lack of systemic disease, CTC analyses and quantitative detection of multi-organ metastases and disseminated tumor cells would need to be undertaken

-At a minimum, authors need to assess lung metastatic burden in vehicle/singly/combo treated mice.

Author Response: We have changed the wording in the abstract and reflected these changes, as suggested by the reviewer, throughout the entirety of the manuscript. Instead of saying "complete response" or "cure", we have said that mice achieved "durable tumor regression".

3. Authors should address why the combination with FEC causes regression in some mice, but the combination of oHSV with AC does not.

Author Response: Authors have addressed this in paragraph #1 of the results (Lines 105-108).

4. Can authors correlate baseline tumor characteristics with eventual combination therapy responses to more clearly understand why some mice respond and others do not? Is this related to perhaps stochastic levels of baseline immune population infiltration in these tumors?

a. Authors could assess immune subpopulation heterogeneity in a cohort of untreated mice to appreciate whether this heterogeneity could explain the rarity of observed regressions.

Author Response: As shown in the survival data (Figures 1 and 2) and IHC analysis (Figure 4 and S4) we see tight clustering of untreated mice (saline), resulting in homogenous tumor growth kinetics. Conversely, the cytokines analysis (Figure 3) shows that untreated mice (saline) have differential expression of some cytokines including G-CSF, GM-CSF, MIP-2 and Fractalkine. Additionally, RNA sequencing (Figure 5a) shows that one mouse clusters differently from the

other 4 untreated (saline) mice. These transcriptomic changes may be due to the nature of subcutaneous tumor models as some tumors inevitably engraft more closely to skin and/or muscle tissue and therefore the TME may be affected by these neighbouring cell types. However, these tumors grown from identical cell populations resulting in slightly different phenotypes at the transcriptomic level is in fact a good clinical representation of the differences we see clinically in TNBC patients. Additional factors such as metabolism, activity and caloric intake will differ between mice and may also contribute to the factors driving response to therapy, but these factors are undoubtedly outside the scope of our work for this particular manuscript.

Reviewer #3 (Remarks to the Author):

Synopsis:

In this manuscript, Vito et al. investigated the effects of low dose chemotherapy combined with oncolytic virotherapy in an E0771 murine TNBC tumor model. They found that the combination therapy increased tumor-infiltrating lymphocytes, contributing to improved responses to checkpoint blockade. The authors demonstrated that the depletion of circulating B cells by anti-CD20 antibodies resulted in the expansion of MDSC and loss of therapeutic efficacy, suggesting that circulating B cells can act as regulators of MDSCs, a key population of cells that drive immune escape and mediate therapeutic resistance.

Criticism:

I think there are some interesting findings in this study: 1) The combination therapy (FEC + oHSV-1) improves overall survival, which is associated with increased tumor-infiltrating lymphocytes. 2) Circulating B cells are required for therapeutic efficacy of the combination therapy. 3) Depletion of circulating B cells affects the immunosuppressive cell populations, including MDSC. Although the finding in #1 has been proposed in many studies [e.g., reviewed in DOI: 10.2147/OV.S66083], the findings #2 and #3 may provide a better understanding of cancer immunotherapy for TNBC. However, as mentioned below, the current dataset appears limited in providing sufficient evidence to support the authors' conclusion. Listed below are the major points/flaws that preclude at this stage acceptance in Communications Biology.

Author Response: The authors would like to thank Reviewer #3 for their time reading and understanding our manuscript. Reviewer #3 has noted the potential clinical translatability of our findings and suggested ways to improve the impact of the manuscript.

Major Points:

1. One of the major concerns of this reviewer is that the immunostimulatory effects of the combination therapy (FEC + oHSV-1) could be potentially due to cancer cell death, not specific to the therapeutic strategy. For instance, is there any difference between the combination therapy and high-dose chemotherapy (that induces similar dose of cancer cell death) in their tumor model, in terms of their effects on cancer immunity? The authors need to confirm that the combination strategy is a key to induce immunogenic response in their TNBC model.

Author Response: In line with many studies in the literature and within our own lab, low dose chemotherapy is well characterized for its immunostimulatory capabilities [1–4]. While high dose chemotherapy may cause temporary tumor regression in hosts, it often leads to tumor relapse and high toxicity profiles. We have indeed done studies in our tumor model (Figure S1) with higher doses of chemotherapy, but mice do not tolerate these treatments well and even mice that initially show responses eventually relapse and rapidly reach tumor endpoint. Additionally, other published findings from our lab also see similar effects, showing the synergism of low dose chemotherapy and oncolytic virotherapy in promoting antitumor immunity [3,5,6].

2. It is not clear whether the therapeutic effect of the combination therapy (“FEC + oHSV-1” and/or “FEC + oHSV-1 + CP”) is mediated by the host immune system. The authors need to use immunocompromised mice to confirm some of their major findings demonstrated in Figures 1 and 2.

Author Response: Given the overwhelming amount of literature supporting the use of low dose chemotherapy and oncolytic virotherapy for immunostimulatory purposes, we would likely not be able to get approval from our animal research ethics board for this study. The major findings found in figures 1 and 2 are further investigated in depth throughout the remainder of the manuscript and we have utilized many different techniques and approaches (in vivo re-challenge study – Figure 2, cytokine analysis – Figure 3, IHC – Figure 4, RNA sequencing – Figure 5, flow cytometry – Figures 7-8) commonly used in the field to assess the immunostimulatory capabilities of our therapeutic platform. Additional to the data in this manuscript, our lab has also shown this to be true in other published articles, consistent with literature in the immunoncology field [3,5–7]. Indeed, the “re-challenge” study (Figure 2D) we performed on mice that had a complete response to treatment is a commonly used study in our field to determine whether or not the host has developed antitumor immunity to a given therapy.

3. The molecular and/or cellular mechanisms of MDSC regulation by circulating B cell is not clear. Because the link between circulating B and MDSC is one of the main findings of this paper, further investigation seems to be necessary.

Author Response: We agree with the reviewer that we have only made preliminary steps towards identifying this phenomenon of B cell regulation of MDSCs and have not shown the exact mechanism of this interaction. However, showing this mechanistically is very technically challenging due to the nature of myeloid cells and difficulties in isolating large, viable numbers of MDSCs for ex vivo functional studies. Investigating the exact mechanism by which this happens would indeed require a great amount of work and characterization and we believe it would increase the impact of this manuscript above the level of Communications Biology publications and likely require a full year of work. We have used our data to put together what we believe is a strong hypothesis of this regulation and have included a new schematic detailing the mechanisms by which we believe our therapy is working (Figure 10, Lines 637-647, shown below). We hope the reviewer will find this schematic useful in detailing the main findings of our work.

Fig. 10. Schematic of therapy-induced immune activation. *Created using BioRender.com.

4. In Figure 4C, the authors showed a significant increase in CD4 T cell infiltration in “FEC + oHSV-1” group. However, in figure S9B, the frequency of CD4 T cells in tumors seems to be unaffected by “FEC + oHSV-1” treatment. Can the author explain this superficial discrepancy? If this can be due to quantification method (histology v.s. FACS), TIL analysis demonstrated in Figure 4 needs to be analyzed again by flow cytometer. Histological analysis is useful to get positional information, but less quantitative.

Author Response: In Figure 4C you can see a rather large spread in levels of CD4⁺ cells in mice treated with FEC + oHSV-1 therapy. If we pay close attention to the p value, it only just qualifies as being statistically significant and can be attributed to the small sample number and dichotomous responses we often see in this treatment group. As we consistently see ~10% of mice that respond to this dual combination therapy, it is logical to say that 1/10 mice will likely show a different TIL profile from the rest of the group. The sample size is much larger in Figure S9B, with the majority of mice having low levels of CD4⁺ cells, and again one mouse is seen to have a much higher level than the others. I suspect that if the n number of this group was smaller it would have also been possible to see a statistical difference. Additionally, histological analysis (as seen in Figure 4) is very good at giving locational information about the amount of infiltration into the core of the tumor of a particular cellular subset. However, as IHC relies on a single tumor slice it does not accurately depict the level of immune cells within an entire tumor mass. For this reason, we chose to show the IHC but further delve into TIL analysis at the end of the manuscript for a deeper assessment of the cell types infiltrating into the entirety of the tumor. We believe the combination of these two modalities gives comprehensive insight into the makeup of our tumors.

Minor Points:

1. Some of the statements are not supported by the demonstrated data. For example, the authors state that “While no monotherapy showed efficacy, the addition of oHSV-1 to the clinical FEC regimen (FEC + oHSV-1) delayed tumor growth in some mice (Figure 1B) and significantly extended survival, with a 10% overall cure rate (Figure 1C)”. To support this statement, the authors need to compare the “FEC” v.s. “FEC + oHSV-1” group, but the p value for this comparison is not indicated in Figure 1C. It would be important to carefully consider each statement.

Author Response: We have added in the p value for this comparison (Lines 121-128, shown below) and changed the wording in the text (Lines 102-105). We have additionally read through the manuscript in its entirety to ensure that all statements have been considered carefully.

A**B****C**
Fig. 1. FEC + oHSV-1 slows tumor growth and decreases tumor kinetics and results in tumor regression in 10% of mice.

(A) C57/Bl6 mice bearing E0771 tumors were treated with saline, chemotherapy (FEC or AC), oncolytic virus (oHSV-1 dICP0) or chemotherapy + oncolytic virus. *Created using BioRender.com. (B) Tumor volumes were measured every 2-3 days from the start of treatment until mice reached endpoint. Each line represents an individual mouse within the group. (C) Kaplan- Meier survival curves of each group. *Mantel-Cox test was used for statistical analyses.

2. It would be important to show the specificity of anti-CD20 antibodies used in this study. For example, the authors may confirm that the populations of other immune cells (e.g., T cell, macrophage,,,) are not affected by anti-CD20 antibody treatment (at the early time point). Or, the authors may use other methods for B cell depletion.

Author Response: Anti-CD20 depletion was confirmed via flow cytometry. In all studies performed, we ensured that we checked a broad spectrum of immune cell populations, though not all subsets were highlighted in the main figures. In Figure S7 we have shown that B cell depletion using anti-CD20 antibody did not cause non-specific depletion of other cellular populations. Indeed, as early as day 6 we can see that CD4⁺ and CD8⁺ cells in mice treated with the anti-CD20 antibody are higher in some treatment groups, and stable in other treatment groups. This also holds true for monocytes and dendritic cells. These findings also hold true in the tumor, spleen and lymph nodes as well (Figure S9), further highlighting the specificity of the antibody chosen.

References

1. Pfirschke, C.; Engblom, C.; Rickelt, S.; Cortez-Retamozo, V.; Garris, C.; Pucci, F.; Yamazaki, T.; Poirier-Colame, V.; Newton, A.; Redouane, Y.; et al. Immunogenic Chemotherapy Sensitizes Tumors to Checkpoint Blockade Therapy. *Immunity* **2016**, doi:10.1016/j.immuni.2015.11.024.
2. Landreneau, J.P.; Shurin, M.R.; Agassandian, M. V.; Keskinov, A.A.; Ma, Y.; Shurin, G. V. Immunological Mechanisms of Low and Ultra-Low Dose Cancer Chemotherapy. *Cancer Microenviron.* **2015**, doi:10.1007/s12307-013-0141-3.
3. Workenhe, S.T.; Pol, J.G.; Lichty, B.D.; Cummings, D.T.; Mossman, K.L. Combining oncolytic HSV-1 with immunogenic cell death-inducing drug mitoxantrone breaks cancer immune tolerance and improves therapeutic efficacy. *Cancer Immunol. Res.* **2013**, doi:10.1158/2326-6066.CIR-13-0059-T.
4. Bracci, L.; Schiavoni, G.; Sistigu, A.; Belardelli, F. Immune-based mechanisms of cytotoxic chemotherapy: Implications for the design of novel and rationale-based combined treatments against cancer. *Cell Death Differ.* 2014.
5. van Vloten, J.P.; Workenhe, S.T.; Wootton, S.K.; Mossman, K.L.; Bridle, B.W. Critical Interactions between Immunogenic Cancer Cell Death, Oncolytic Viruses, and the Immune System Define the Rational Design of Combination Immunotherapies. *J. Immunol.* **2018**, doi:10.4049/jimmunol.1701021.
6. Workenhe, S.T.; Nguyen, A.; Bakhshinyan, D.; Wei, J.; Hare, D.N.; MacNeill, K.L.; Wan, Y.; Oberst, A.; Bramson, J.L.; Nasir, J.A.; et al. De novo necroptosis creates an inflammatory environment mediating tumor susceptibility to immune checkpoint inhibitors. *Commun. Biol.* **2020**, doi:10.1038/s42003-020-01362-w.
7. Cuddington, B.P.; Mossman, K.L. Oncolytic bovine herpesvirus type 1 as a broad spectrum cancer therapeutic. *Curr. Opin. Virol.* 2015.

Reviewers' Comments:

Reviewer #3:

Remarks to the Author:

Vito and colleagues have improved their manuscript based on my prior critiques. The addition of patient cohort RNA seq data as well as validation of results in the PY230 model have substantially improved the rigor of the experimental findings. It would be beneficial to speculate about the inter-mouse heterogeneity of responses in the discussion section. Otherwise, my comments have been satisfactorily addressed.

Reviewer #4:

Remarks to the Author:

Given that the link between circulating B and MDSC is one of the main conceptual advance of this paper, this reviewer still thinks that it is important to investigate molecular and/or cellular mechanisms of MDSC regulation by B cells. However, I respect the authors' view. The authors have satisfactorily responded to other questions and made the necessary changes to the manuscript.

Reviewer #2 (Remarks to the Author):

Vito and colleagues have improved their manuscript based on my prior critiques. The addition of patient cohort RNA seq data as well as validation of results in the PY230 model have substantially improved the rigor of the experimental findings. It would be beneficial to speculate about the inter-mouse heterogeneity of responses in the discussion section. Otherwise, my comments have been satisfactorily addressed.

Response from the Author:

We would like to thank reviewer #2 for their help and guidance in greatly improving our manuscript. The final request about inter-mouse heterogeneity has been addressed in the final paragraph of the discussion (Lines 398-402), as follows:

“It is common to have patients with phenotypically similar cancers present with varying response levels to treatment. This phenomenon was also seen in our studies, in which mice with identical genetic makeup had dichotomous responses to the same treatment. This inter-mouse heterogeneity may be due to the individual metabolic activity and immune profile of each individual mouse.”